# Lack of MDA5 delays hematopoietic aging by modulating inflammaging and proteostasis in mice

Veronica Bergo[1,2,3,11] ✉, Pavlos Bousounis[1,2,11], Giang To Vu[4,11], Mélodie Douté [5], Aikaterini Polyzou[1,2,4], Maria-Eleni Lalioti[6], Bogdan B. Grigorash [4], Lyudmila Tsurkan[5], Nicholas Morchel[5], Ward Deboutte [1], Frédéric Brau[4], Thomas Manke [1,7], Sagar [8], Hind Medyouf [9,10], Dmitry V. Bulavin[4], Nina Cabezas-Wallscheid[6], Marta Derecka [5] ✉ & Eirini Trompouki [1,4] ✉

"Inflammaging", the chronic increase in inflammatory signaling with age, remains poorly understood in hematopoietic aging. Here, we identify the innate immune RNA sensor melanoma differentiation–associated protein 5 (MDA5) as an important factor of hematopoietic stem cell (HSC) aging. Aged *Mda5*[-/-] mice exhibit reduced HSC accumulation and myeloid bias. Importantly, aged *Mda5*[-/-] HSCs retain greater quiescence and superior repopulation capacity in noncompetitive transplants compared to wild-type counterparts. Multiomic analyses– including chromatin accessibility, transcriptomics, and metabolomics–reveal decreased inflammatory signaling, a youthful metabolic profile, and improved proteostasis in *Mda5*[-/-] HSCs, through regulation of HSF1 and phospho-EIF2A, key proteostasis regulators. Activation of HSF1 in aged wild-type HSCs partially restores youthful features, supporting a causal role for proteostasis maintenance. Collectively, our findings demonstrate that attenuating MDA5-dependent inflammation preserves HSC function during aging by maintaining metabolic fitness and proteostasis and provide insight into potential therapeutic strategies for mitigating hematopoietic aging.

Hematopoietic stem cells (HSC) lie at the top of the hematopoietic hierarchy and give rise to most of the differentiated hematopoietic cells. HSCs are normally quiescent within the bone marrow and can only be activated by external stimuli to exit quiescence and enter the cell cycle[1]. Inflammation holds a prominent role throughout the lifespan of HSCs. Indeed, it has been shown that IFNα (interferon a), IFNγ, TNF (tumor necrosis factor), and IL-1β (interleukin 1β) are responsible for the exit of HSCs from quiescence[2–6]. IL-1 mediates microbiome-induced inflammaging of HSCs in mice[7]. It has been shown that inflammatory signaling promotes clonal expansion in cases of clonal hematopoiesis (CH)[8,9]. At the same time, it was recently proposed that the mutant clones are resistant to the detrimental effects of the

[1]Max Planck Institute of Immunobiology and Epigenetics, Freiburg, Germany. [2]Faculty of Biology, University of Freiburg, Freiburg, Germany. [3]International Max Planck Research School for Immunobiology, Epigenetics and Metabolism (IMPRS-IEM), Freiburg, Germany. [4]Institute for Research on Cancer and Aging of Nice (IRCAN), CNRS, INSERM, Université Côte d'Azur, Nice, France. [5]Department of Hematology, St. Jude Children's Research Hospital, Memphis, TN, USA. [6]Department of Health Sciences and Technology, Laboratory of Stem Cell Biology and Ageing, Swiss Federal Institute of Technology (ETH Zürich), Zürich, Switzerland. [7]HTW Berlin, School of Computing, Communication and Business, Berlin, Germany. [8]Department of Medicine II, Gastroenterology, Hepatology, Endocrinology and Infectious Diseases, Faculty of Medicine, Freiburg University Medical Center, University of Freiburg, Freiburg, Germany. [9]Department of Hematology, Oncology, Hemostaseology, and Stem Cell Transplantation, Faculty of Medicine, RWTH Aachen University, Aachen, Germany. [10]Center for Integrated Oncology Aachen Bonn Cologne Düsseldorf (CIO ABCD), Düsseldorf, Germany. [11]These authors contributed equally: Veronica Bergo, Pavlos Bousounis, Giang To Vu. ✉e-mail: bergo95veronica@gmail.com; Marta.Derecka@stjude.org; Eirini.TROMPOUKI@univ-cotedazur.fr

inflammatory milieu, thus leading to their expansion[10]. During aging, several features of HSCs become compromised. Aged HSCs exhibit reduced self-renewal and repopulation capacity, a myeloid bias, and an increase in their frequency, while proteostatic mechanisms are compromised concomitant with chronic low-grade inflammation[11–13]. "Inflammaging" refers to the low-grade, sterile chronic inflammation affecting the aged hematopoietic system and HSC function, including metabolic capacity, differentiation, and self-renewal capacity in aged organisms, leading to various diseases[11,12,14]. Indeed, it has been shown that continuous exposure to inflammation in the form of the viral mimic PolyI:C leads to impairment of HSC self-renewal capacity and premature aging[15]. However, how inflammation drives hematopoietic aging and the mechanisms through which the two phenomena are linked remain understudied.

An inflammatory pathway whose role in "inflammaging" has not been studied before is the RIG-I-like receptor (RLR) pathway[16]. RLRs are a group of three intracellular innate immune receptors: RIG-I (retinoic acid-inducible gene I) like receptor, MDA5 (melanoma differentiation-associated protein 5), and LGP2 (laboratory of genetics and physiology 2). Upon viral RNA recognition, RIG-I and MDA5 form filaments and interact with the downstream adapter protein MAVS (mitochondrial antiviral signaling protein), thus creating a signaling platform that permits the recruitment and activation of multiple signaling molecules. Ultimately, activation of the RLR signaling pathway leads to induction of a type-I IFN response and the secretion of proinflammatory cytokines[16,17]. We have previously shown that RLRs play a key role in hematopoietic stem and progenitor cell (HSPC) formation during development[18], and that MDA5 modulates adult HSC activation after chemotherapy[19]. Whether RLRs are involved in hematopoietic aging remains unknown. Recently, it was reported by us and others that not only viral RNA, but also other types of RNA, including mitochondrial and transposable element RNA, can also be a stimulus for RLR receptors[20–25]. Since transposable element expression in HSCs increases with age[26], MDA5 may play a role in HSC aging.

Protein homeostasis, or proteostasis, encompasses all the processes that work together to maintain the proper levels, structure, and function of proteins within living organisms[27]. Although inflammaging and impaired proteostasis are both hallmarks of aging HSCs, a link between them remains unexplored. Young adult HSCs have lower protein synthesis rates compared to downstream progenitors[28,29]. Heat shock transcription factor 1 (HSF1) is a master regulator of proteostasis[30–32] which was recently reported to promote proteostasis and the regenerative activity of HSCs in response to culture stress and aging[33]. HSF1 protein levels decline with age in many tissues[34]. Protein translation inhibition through PKR (protein kinase R) or PERK (protein kinase R-like endoplasmic reticulum kinase)-induced phospho-EIF2a (eukaryotic translation initiation factor 2a) is also important for proteostasis and the induction of autophagy[35]. Autophagy helps to remove cytosolic protein aggregates and damaged organelles inside cells[36]. Loss of autophagy in HSCs leads to metabolic stress and altered functionality. While most HSCs lose autophagic capacity with age, a fraction maintains high autophagy and resembles young HSCs[37]. Chronic low-grade inflammation is the signal that leads to autophagy maintenance through metabolic adaptations[38]. Finally, aggrephagy has been shown as the preferred method of protein degradation by HSCs[39].

Here, we show that, in the absence of MDA5, HSCs are protected from age-associated functional decline. Loss of MDA5 preserves stem cell quiescence, limits myeloid bias, and maintains repopulation capacity during aging by restraining inflammatory signaling and sustaining metabolic fitness and proteostasis. These findings establish innate RNA sensing as an intrinsic driver of hematopoietic stem cell aging and highlight modulation of MDA5-dependent pathways as a potential strategy to mitigate age-related hematopoietic dysfunction.

## Results

### Reduced inflammation in *Mda5*[-/-] HSCs and bone marrow serum

Since activation of MDA5 induces a type-I IFN response and the secretion of pro-inflammatory cytokines, we isolated bone marrow serum from aged (18 to 24 months old) wild-type WT (or *Mda5*[+/+] hereafter WT) and *Ifih1*-knockout mice (also known as *Mda5*; hereafter *Mda5*[-/-]; *B6.Cg-Ifih1tm1.1Cln/J*)[40], to measure secreted cytokines. Indeed, IFN-β concentration was significantly lower in the bone marrow of aged *Mda5*[-/-] mice (Fig. 1A). Concomitantly, other cytokines such as IL-1α and β, IL-10, IL-6, and others exhibited lower concentrations in *Mda5*[-/-] bone marrow serum, while other cytokines like TNF and IFNγ showed no differences (Fig. 1B and Supplementary Fig. 1A, B). We repeated the experiment with young (2–3 months old) and middle-aged (10–14 months old) mice and observed various cytokines that had lower concentrations in the bone marrow serum of young and middle-aged *Mda5*[-/-] animals compared to controls (Supplementary Fig. 1C, D). When comparing these results at different ages, we noticed that cytokine concentration was increasing with age, with IL-1α, IL-10, IFNβ, and TNF being the most highly secreted in the bone marrow. This effect was dampened in aged *Mda5*[-/-] bone marrow serum (Fig. 1C).

We then wanted to check whether chromatin accessibility for inflammatory motifs is different in *Mda5*[-/-] HSCs. To explore the chromatin accessibility landscape of aged WT and *Mda5*[-/-] HSCs, we performed an assay for transposase-accessible chromatin followed by sequencing (ATAC-seq). We isolated HSCs using EPCR (endothelial protein C receptor) SLAM markers (EPCR + CD150 + CD48−), as EPCR has been proven to be a stable marker that is insensitive to stress signals[41,42]. Indeed, more than 97% of the EPCR SLAM HSCs are also Lin-cKit + Sca+ (LSK) CD150 + CD48− (LSK SLAM) (Supplementary Fig. 1E). Similar results were obtained for middle-aged and young HSCs (Supplementary Fig. 1F, G). Therefore, in this text, "HSC" will henceforth refer to EPCR SLAM HSCs, unless stated otherwise. Clustering of samples by age was observed (Supplementary Fig. 1H). We compared the accessible chromatin regions between young or aged WT and *Mda5*[-/-] HSCs. In the comparison of the young WT and *Mda5*[-/-] HSCs, we found 1.723 regions that were mostly accessible in WT (reduced peaks), and 10.694 mostly accessible in *Mda5*[-/-] HSCs (induced peaks) (Fig. 1D and Supplementary Data 1–4). Aged *Mda5*[-/-] HSCs on the other hand have less accessible chromatin than the WT, with 11.904 sites induced in WT and 5.336 sites induced in *Mda5*[-/-] HSCs (Fig. 1E and Supplementary Data 1–4). Both induced and reduced peaks were distributed in similar proportions across transcription start sites (TSS), exons, introns, and intergenic regions (Fig. 1D, E). Only the reduced peaks in young Mda5[-/-] HSCs were found mostly around the TSS (Fig. 1D). Additionally, when we compared young and aged WT or *Mda5*[-/-] HSCs, we observed an opening of chromatin in the aged populations that was distributed similarly in different genomic regions (Fig. 1F, G). Comparison of young and aged samples together revealed that most of the common induced regions were between the aged WT and *Mda5*[-/-] HSCs, and thus the aged *Mda5*[-/-] HSCs do not resemble younger WT HSCs (Fig. 1H).

Known motif analysis revealed that inflammatory motifs like IRF3 (IFN regulatory factor 3) were mainly enriched in aged *Mda5*[-/-] reduced peaks (*p*-value 1e-21 in reduced and 1e-05 in induced sites), but not young WT or *Mda5*[-/-] regions (Fig. 1I). In detail, in the young WT versus *Mda5*[-/-] regions, we found NFY (nuclear transcription factor Y,1e-49), KLF1 (Krueppel-like factor 1, 1e-47), SP1 (specificity protein 1, 1e-38), FLI1 (friend leukemia integration 1,1e-21) in the reduced regions and ZSCAN1 (zinc finger and SCAN domain containing 1), P53 (both 1e-02) in the induced regions (Fig. 1I). In aged WT versus *Mda5*[-/-] regions we found CTCF (CCCTC-binding factor, 1e-911), ETV2 (Ets variant 2, 1e-248), ERG (ETS transcription factor ERG, 1e-232), IRF3 (1e-21), PU1 (1e-133), JUN-AP1(1e-72) in the reduced regions and CTCF (1e-139), ETV2 (1e-69), ETS1 (1e-55), PU1 (1-36), RUNX (1e-31), IRF3 (1e-05) in the induced regions (Fig. 1I). Thus, the inflammatory IRF3 motif is less

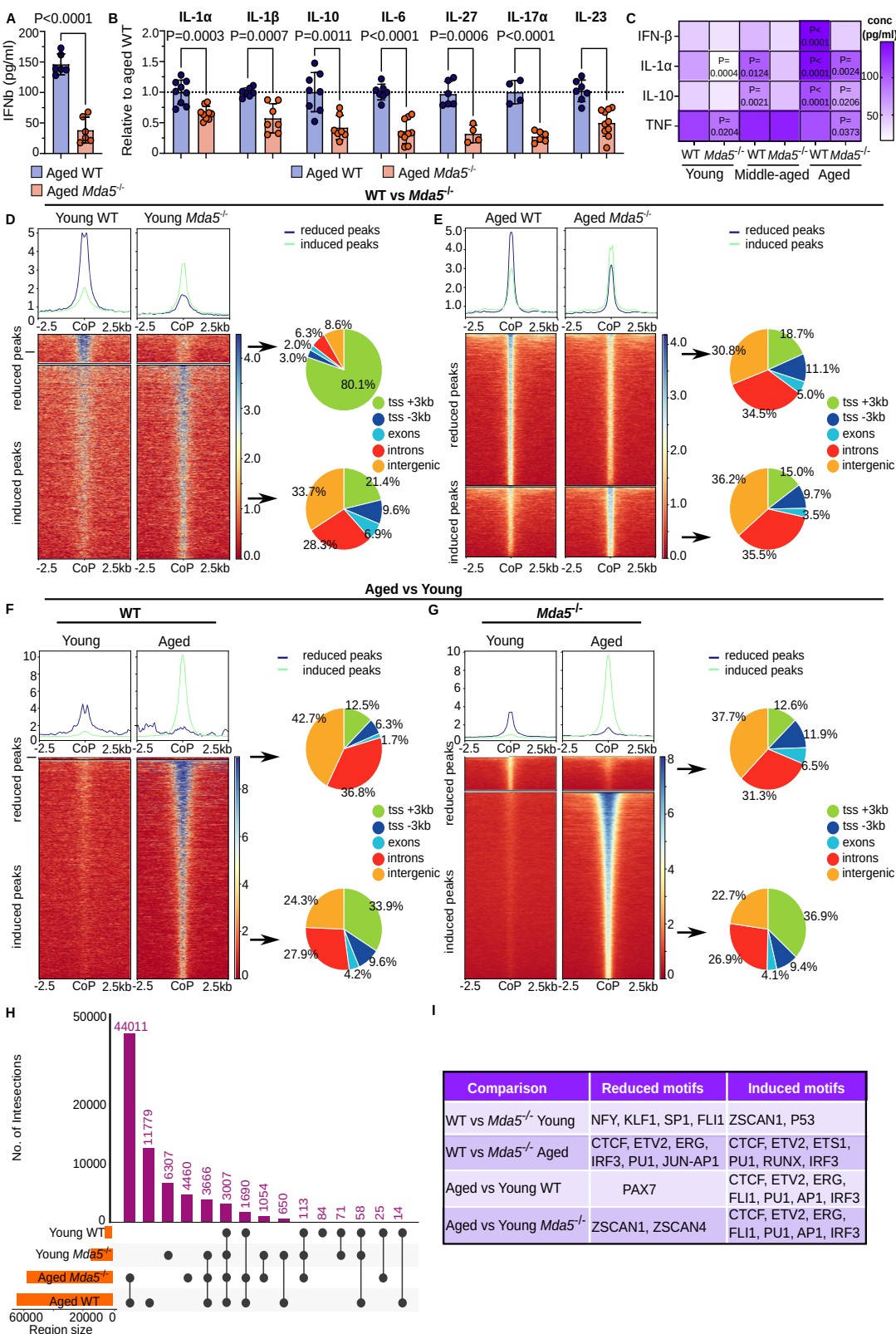

present in induced *Mda5⁻/⁻* regions. The young WT HSCs harbor motifs for Pax7 (1e-2) while the induced regions in the aged WT HSCs harbor CTCF (1e-1965), ETV2 (1e-1063), ERG (1e-1032), FLI1 (1e-993), PU1 (1e-577), AP1 (activating protein-1, 1e-300), and IRF3 motifs (1e-131) (Fig. 1I). In contrast young *Mda5⁻/⁻* HSCs harbor ZSCAN (1e-02) motifs while for the induced regions in aged *Mda5⁻/⁻* HSCs we observed CTCF (1e-1177),

ETV2 (1e-1057), ERG (1e-1003), FLI1 (1e-1015), PU1 (1e-552), AP1 (1e-211), IRF3 (1e-86) motifs (Fig. 1I).

We performed digital footprinting analysis to identify motif occupancy for transcription factors IRF3, NF-κB (p65), and STAT1 in aged compared to young WT and *Mda5⁻/⁻* HSCs. We observed a substantial increase in occupancy between young and aged WT or

**Fig. 1 | Reduced inflammation in *Mda5*<sup>-/-</sup> HSCs and bone marrow serum. A** IFNβ concentration (pg/ml) in the bone marrow serum of aged WT or *Mda5*<sup>-/-</sup> mice. *N* = 6 biologically independent samples in *n* = 2 independent experiment. Each dot represents one mouse. Data are presented as mean values ± SD. Two-tailed unpaired t-test (*P* = 0.000002). **B** Fold change of cytokine concentrations in the bone marrow serum of aged WT or *Mda5*<sup>-/-</sup>, normalized to the corresponding WT control. *N* = 9 WT or *Mda5*<sup>-/-</sup> for IL-1α, *N* = 6 WT or *N* = 7 *Mda5*<sup>-/-</sup> for IL-1β, *N* = 8 WT or *N* = 7 *Mda5*<sup>-/-</sup> for IL-10, *N* = 7 WT or *N* = 9 *Mda5*<sup>-/-</sup> for IL-6, *N* = 5 WT or *N* = 4 *Mda5*<sup>-/-</sup> for IL-27, *N* = 4 WT or *N* = 6 *Mda5*<sup>-/-</sup> for IL-17α, *N* = 7 WT or *N* = 10 *Mda5*<sup>-/-</sup> for IL-23 biologically independent samples in *n* = 4 independent experiments. Each dot represents one mouse. Data are presented as mean values ± SD. Two-tailed unpaired t-tests, *P* (IL-6 = 0.000004; *P* (IL-17a) = 0.000032; *P* (IL-23) = 0.000099. **C** Cytokine concentration in bone marrow serum of young (*N* = 3 WT and 5 *Mda5*<sup>-/-</sup> for IFNβ, *N* = 3 WT and 5 *Mda5*<sup>-/-</sup> for IL-1α, *N* = 4 WT and 3 *Mda5*<sup>-/-</sup> for IL-10, *N* = 3 WT and 4

*Mda5*<sup>-/-</sup> for TNF), middle-aged (*N* = 4 WT and 3 *Mda5*<sup>-/-</sup> for IFNβ, *N* = 2 WT and 3 *Mda5*<sup>-/-</sup> for IL-1α, *N* = 3 WT and 3 *Mda5*<sup>-/-</sup> for IL-10, *N* = 4 WT and 3 *Mda5*<sup>-/-</sup> for TNF), and aged (*N* = 5 WT and 5 *Mda5*<sup>-/-</sup> for IFNβ, *N* = 5 WT and 4 *Mda5*<sup>-/-</sup> for IL-1α, *N* = 4 WT and 4 *Mda5*<sup>-/-</sup> for IL-10, *N* = 3 WT and 2 *Mda5*<sup>-/-</sup> for TNF), WT or *Mda5*<sup>-/-</sup> mice biologically independent samples in *n* = 2 independent experiment. Data are presented as mean values. Two-way ANOVA, normalized to young WT. Heatmaps (left) of the differentially accessible regions in young WT and young *Mda5*<sup>-/-</sup> HSCs (**D**), aged WT and aged *Mda5*<sup>-/-</sup> HSCs (**E**), WT young versus aged (**F**) and *Mda5*<sup>-/-</sup> young versus aged (**G**) ±2, 5 kb from the center of the peak (CoP). Pie charts representing the genomic distribution (right) of gained and lost regions in (**D–G**). TSS: transcriptional start site. **H** Upset plot showing the number of intersections between narrow peaks from young WT, young *Mda5*<sup>-/-</sup>, aged WT and aged *Mda5*<sup>-/-</sup>, HSCs. **I** Known motif analysis in induced or reduced regions of chromatin accessibility of the indicated comparisons.

---

*Mda5*<sup>-/-</sup> footprints, although the total number of footprints was lower in the knockout. Additionally, we did not detect many footprints for inflammatory factors in young HSCs (Supplementary Fig. 1I). To further investigate this, we compared aged WT and *Mda5*<sup>-/-</sup> footprints, and we observed less occupancy of the WT footprints in the *Mda5*<sup>-/-</sup> samples (Supplementary Fig. 1J).

Taken together, these results further support that inflammatory signaling activation, a hallmark of aging, is significantly dampened in *Mda5*<sup>-/-</sup> HSCs and bone marrow.

### Reduced HSC accumulation and myeloid bias in aged *Mda5*<sup>-/-</sup> animals

We reasoned that changes in cytokine concentrations and inflammatory signaling activation could impact HSC functionality and maintenance during aging. We did not detect changes in bone marrow cellularity in young, middle-aged, or aged WT and *Mda5*<sup>-/-</sup> animals (Supplementary Fig. 2A). When evaluating the HSC compartment in middle-aged and aged WT animals, we observed an increased frequency of EPCR SLAM, LSK SLAM, and LT-HSCs, indicative of HSC expansion, an established hallmark of aging, that was more prominent in aged compared to the young or middle-aged animals. However, this accumulation was significantly reduced in *Mda5*<sup>-/-</sup> animals (Fig. 2A–C and Supplementary Fig. 2B–D). Concomitantly, metabolomic analysis showed significant differences between aged WT and *Mda5*<sup>-/-</sup> HSCs. The concentration of glutathione disulfide (GSSG), which is known to increase during aging[43], was lower in the aged *Mda5*<sup>-/-</sup> HSCs (Fig. 2D). The nicotinamide adenine dinucleotide (NAD<sup>+</sup>) concentration has been shown to decline upon aging. Similarly, a decline in NADP<sup>+</sup> is also expected during aging, since it is exclusively synthesized from NAD<sup>+</sup> by cytoplasmic and mitochondrial NAD<sup>+</sup> kinases[44]. The concentration of these two metabolites, and especially NADP<sup>+</sup>, was higher in aged *Mda5*<sup>-/-</sup> HSCs (Fig. 2D). Minimal differences were observed in the nucleotide metabolism and the TCA cycle between aged *Mda5*<sup>-/-</sup> and WT HSCs (Supplementary Fig. 2E, F). Additionally, we measured HSC mitochondrial mass and membrane potential using MitoTracker green dye (MTG) and tetramethylrhodamine methyl ester (TMRM), respectively, in the presence of verapamil[45,46], a known blocker of xenobiotic efflux pumps, which fosters dye retention in the organelles. This allowed us to observe a higher mitochondrial mass and membrane potential in aged *Mda5*<sup>-/-</sup> HSCs, a characteristic of young HSCs[47–49], albeit in studies utilizing verapamil (Fig. 2E, F).

Next, we evaluated the MPP (multipotent progenitor) composition in the bone marrow of young, middle-aged and aged mice. We observed an expansion of the MPP1 population in both aged WT and *Mda5*<sup>-/-</sup> animals, but to a lesser degree in the aged *Mda5*<sup>-/-</sup> (Supplementary Fig. 2G). MPP2 cells also accumulated with age, to a similar extent between WT and *Mda5*<sup>-/-</sup> animals (Supplementary Fig. 2H). The frequency of MPP3 cells, which represent myeloid-biased progenitor cells[50], increased in aged WT mice compared to aged *Mda5*<sup>-/-</sup> animals (Supplementary Fig. 2I). Lastly, the MPP4 population, comprised of the

lymphoid-prone progenitor cells[50], equally declined with age in WT and *Mda5*<sup>-/-</sup> animals (Supplementary Fig. 2J). Taken together, these results indicate a myeloid bias within the WT progenitor compartment during aging, which is ameliorated by *Mda5* deficiency.

We also assessed the frequency of committed progenitors, including common myeloid progenitors (CMP), granulocytic monocytic progenitors (GMP), megakaryocytic-erythroid progenitors (MEP) and common lymphoid progenitors (CLP), as well as differentiated lineage populations in the bone marrow. While all the population frequencies were comparable in young animals, CMPs and MEPs were decreased while CLPs were increased in middle-aged and aged *Mda5*<sup>-/-</sup> compared to age-matched WT mice (Fig. 2G–L).

In conclusion, HSCs tend to accumulate less, and there is less myeloid bias during aging in *Mda5*<sup>-/-</sup> animals.

### Aged *Mda5*<sup>-/-</sup> HSCs are more quiescent and have better repopulation capacity

To functionally investigate the effects of the lack of *Mda5* in aging, we first asked whether their quiescent status differs from that of the aged WT HSCs. We isolated HSCs from young, middle-aged, and aged WT and *Mda5*<sup>-/-</sup> animals, and we counted the number of cell divisions after 24 and 48 h in culture. The percentage of WT HSCs undergoing at least one division increased during aging, but overall, *Mda5*<sup>-/-</sup> HSCs divided less than their WT counterparts at all ages (Fig. 3A and Supplementary Fig. 3A). Similarly, more *Mda5*<sup>-/-</sup> HSCs remained in G0 phase at all ages when comparing the cell cycle status of the WT to the *Mda5*<sup>-/-</sup> HSCs (Fig. 3B and Supplementary Fig. 3B).

Next, we assessed the repopulation capacity in non-competitive transplantations. In these experiments, we transplanted the volume of whole bone marrow containing 300 HSCs from aged WT or *Mda5*<sup>-/-</sup> animals (CD45.2) into lethally irradiated recipients (CD45.1). The overall chimerism of *Mda5*<sup>-/-</sup> HSCs was significantly higher compared to their WT counterparts, both in primary and secondary transplantations (Fig. 3C–J). However, this reconstitution advantage was lost in competitive transplantations, where the volume of whole bone marrow containing 300 HSCs of either aged WT or *Mda5*<sup>-/-</sup> animals (CD45.2) was transplanted with 0.5 million bone marrow cells from a young donor (CD45.1) into lethally irradiated recipients (CD45.1/.2) (Supplementary Fig. 3C–J). Thus, aged *Mda5*<sup>-/-</sup> HSCs have better repopulation capacity in a non-competitive setting, but they lose this advantage when competing with young HSCs.

To examine the capacity of WT and *Mda5*<sup>-/-</sup> HSCs to repopulate lethally irradiated recipients, we sorted and transplanted 100 LSK-SLAM HSCs from young or aged WT or *Mda5*<sup>-/-</sup> mice together with aged-matched CD45.1 competitors into CD45.1 lethally irradiated recipients, followed by whole bone marrow competitive secondary transplantations. Overall, we observed similar total chimerism in the primary competitive transplantations of young and aged WT and *Mda5*<sup>-/-</sup> HSCs (Fig. 3K–O and Supplementary Fig. 3K–O). However, the chimerism of young *Mda5*<sup>-/-</sup> HSCs was higher in the secondary

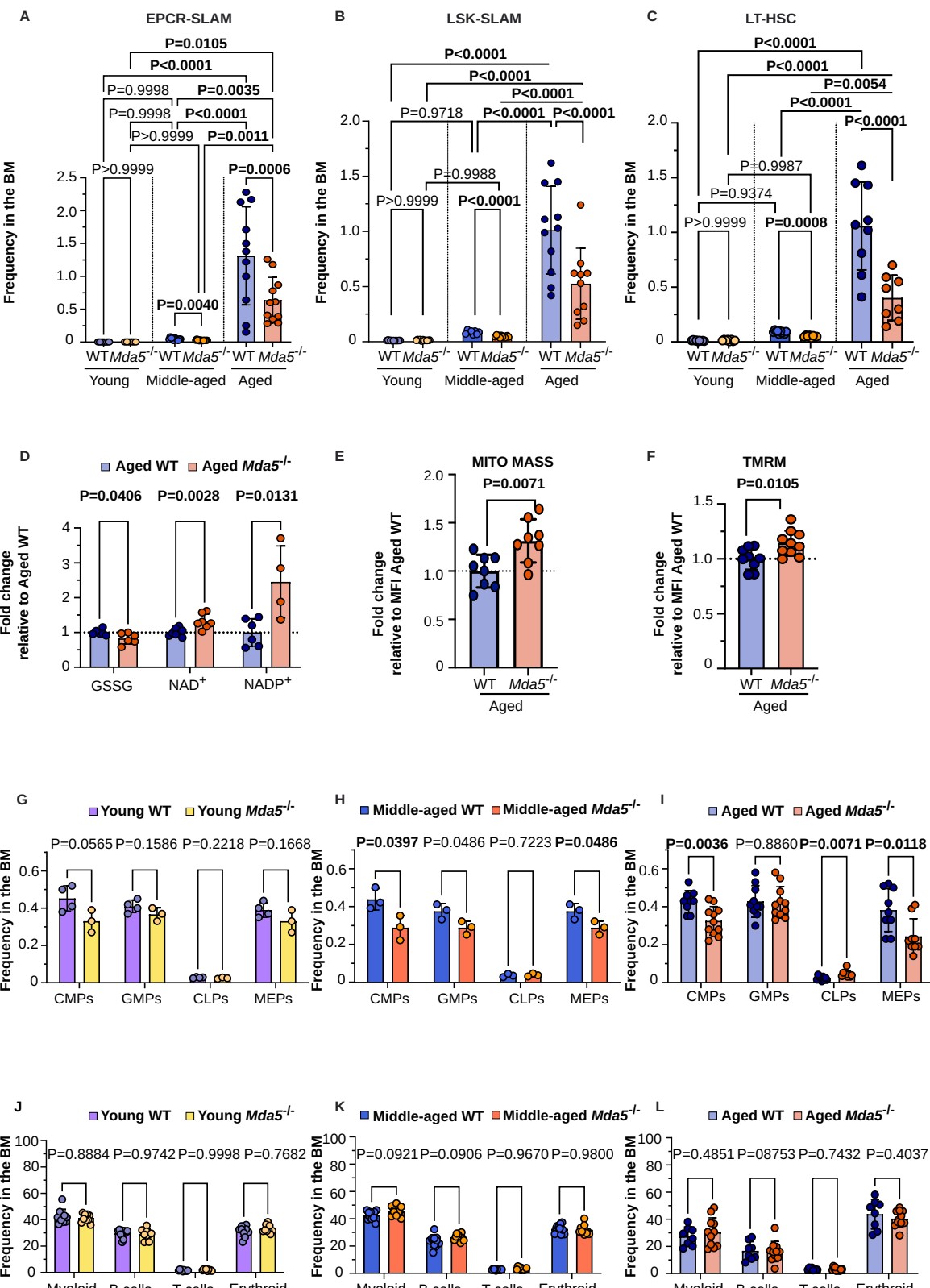

competitive transplantations when compared to young WT HSCs (Supplementary Fig. 3K), with a significantly higher contribution of CD45.2 donor-derived cells to B cell, T cell, monocyte, and granulocyte populations (Supplementary Fig. 3L–O). One potential explanation for the significantly increased chimerism of the secondary transplants of the young *Mda5*⁻ is the increase in the frequency of the LSKs and HSCs in the primary transplant (not significant due to the transplanted

animals with very low chimerism). In contrast, the total and lineage-specific chimerism of aged *Mda5*⁻ HSCs was comparable to the WT counterpart (Fig. 3K–P).

In line with these results, analysis of the bone marrow of primary recipients revealed no significant differences between WT and *Mda5*⁻ in the overall chimerism and the chimerism of lineage-specific and hematopoietic progenitor populations (Supplementary Fig. 4A, C).

**Fig. 2 | Reduced HSC accumulation and myeloid bias in aged *Mda5*[-/-] animals.** Frequency of EPCR SLAM (**A**), LSK SLAM (**B**), and LT-HSC (**C**) populations in young (*N* = 7 WT and 6 *Mda5*[-/-] for EPCR SLAM, *N* = 11 WT and 11 *Mda5*[-/-] for LSK SLAM, *N* = 9 WT and 8 *Mda5*[-/-] for LT HSCs) middle-aged (*N* = 9 WT and 11 *Mda5*[-/-] for EPCR SLAM, *N* = 8 WT and 12 *Mda5*[-/-] for LSK SLAM, *N* = 10 WT and 8 *Mda5*[-/-] for LT HSCs) and aged (*N* = 11 WT and 11 *Mda5*[-/-] for EPCR SLAM, *N* = 11 WT and 10 *Mda5*[-/-] for LSK SLAM WT, *N* = 9 WT and 8 *Mda5*[-/-] for LT HSCs) biologically independent samples in *n* = 3 independent experiments. Each dot represents one mouse. Data are presented as mean values ± SD. One-way ANOVA. For **A**: *P* (middle-aged WT vs. aged WT) = 0.0000000004; *P* (young WT vs. middle-aged WT) = 0.9997836098; *P* (young WT vs. aged WT) = 0.0000000055; *P* (young *Mda5*[-/-] vs. middle-aged *Mda5*[-/-]) = 0.9999911871. For **B**: *P* (young WT vs. young *Mda5*[-/-]) = 0.99999999475; *P* (middle-aged WT vs. middle-aged *Mda5*[-/-]) = 0.00000351753; *P* (middle-aged WT vs. aged WT) = 0.00000000001; *P* (young WT vs. aged WT) = 0.00000000001; *P* (aged WT vs. aged *Mda5*[-/-]) = 0.00002793280; *P* (middle-aged *Mda5*[-/-] vs. aged *Mda5*[-/-]) = 0.00002727944; *P* (young *Mda5*[-/-] vs. aged *Mda5*[-/-]) = 0.00000992713. For **C**: *P* (young WT vs. young *Mda5*[-/-]) = 0.999999997476817; *P* (aged WT vs. aged *Mda5*[-/-]) = 0.000000053123669; *P* (middle-aged WT vs. aged WT) < 0.000000000000001; *P* (young WT vs. aged WT) < 0.000000000000001; *P* (young *Mda5*[-/-] vs. aged *Mda5*[-/-]) < 0.000000000000001. **D** Fold change of intracellular redox-metabolite concentrations of aged WT or *Mda5*[-/-] HSCs, relative to WT control. *N* = 6 WT and *N* = 6 *Mda5*[-/-] for GSSG, *N* = 8 WT and *N* = 7 *Mda5*[-/-] for NAD+, *N* = 6 WT and *N* = 4 *Mda5*[-/-] for NAPD+ biologically independent samples in *n* = 3 independent experiments. Data are presented as mean values ± SD. Two-tailed unpaired t-tests ± SD. Relative mitochondrial mass (**E**) and membrane potential (**F**) of aged WT or *Mda5*[-/-] HSCs. *N* = 8 WT and *N* = 8 *Mda5*[-/-] for (**E**), *N* = 9 WT and *N* = 10 for (**F**) biologically independent samples in *n* = 3 independent experiments. Each dot represents one mouse. Data are presented as mean values ± SD. Two-tailed unpaired t-test. Frequency of committed progenitor populations in the bone marrow of young WT or *Mda5*[-/-] mice (**G** *N* = 4 WT and 3 *Mda5*[-/-] biologically independent samples in *n* = 1 independent experiments, middle-aged WT or *Mda5*[-/-] mice (**H** *N* = 3 WT and 3 *Mda5*[-/-] biologically independent samples in *n* = 1 independent experiments) or aged WT or *Mda5*[-/-] mice (**I** *N* = 9 WT and 11 *Mda5*[-/-] for CMPs, *N* = 10 WT and 11 *Mda5*[-/-] for GMPs, *N* = 9 WT and 9 *Mda5*[-/-] for CLPs and MEPs biologically independent samples in *n* = 3 independent experiments). CMPs common myeloid progenitors, CLPs common lymphoid progenitors, GMPs granulocyte-macrophages progenitors, MEPs megakaryocytic-erythroid progenitors. Each dot represents one mouse. Data are presented as mean values ± SD. Two-tailed unpaired t-tests. Lineage population distribution in the bone marrow of young (**J** *N* = 10 WT and 9 *Mda5*[-/-]), middle-aged (**K** *N* = 11) aged (**L** *N* = 8 WT and 11 *Mda5*[-/-] biologically independent samples in *n* = 3 independent experiments.). Each dot represents one mouse. Data are presented as mean values ± SD. Two-tailed unpaired t-tests.

Yet, the contribution of CD45.2 donor-derived cells in the bone marrow of secondary recipients was higher for young and aged *Mda5*[-/-] HSCs (Supplementary Fig. 4B, D). This confirms that both have better reconstitution capacity upon secondary transplantations and indicates that *Mda5*[-/-] HSCs show diminished differentiation, potentially due to their more quiescent nature.

Taken together, these results show that aged *Mda5*[-/-] are more quiescent and have a better reconstitution capacity in a non-competitive setting, as well as in the bone marrow of secondary competitive transplantations.

## Proteostatic pathways are deregulated in middle-aged and aged *Mda5*[-/-] HSCs

To investigate the molecular mechanism underlying the youthful phenotype of *Mda5*[-/-] HSCs, we performed bulk RNA-sequencing (RNA-seq) to compare young, middle-aged, and aged WT and *Mda5*[-/-] HSC transcriptomes. Multiple genes were deregulated (fold change 1.5 and FDR < 0.05) in all comparisons, but only *ifih1* was significantly deregulated in the comparison between young and *ifih1* and another gene were significantly deregulated in middle-aged WT and *Mda5*[-/-] HSCs (Supplementary Data Set 5). Concomitantly, principal component analysis (PCA) showed a clear separation of the WT and *Mda5*[-/-] HSC aged transcriptomes from the other transcriptomes, while *Mda5*[-/-] middle-aged and young HSC transcriptomes clustered together with the WT (Fig. 4A). In detail, one WT and one *Mda5*[-/-] young transcriptome clustered closer to the aged transcriptomes (Fig. 4A). We next depicted the top 500 most differentially expressed genes clustered by age, where the aged *Mda5*[-/-] HSCs clustered next to the young *Mda5*[-/-] HSCs (Fig. 4A). Of note, *Ifih1* alias *Mda5* was not transcriptionally upregulated during aging in our datasets (Supplementary Data 5) nor in an examined, publicly available, human dataset[51]. Gene set enrichment analysis (GSEA) showed enrichment for the dormant HSC signature[52,53] in young, middle-aged, and aged *Mda5*[-/-] HSCs in comparison to the respective WT (Fig. 4B–D), verifying our single-cell division and cell cycle assays. Additionally, the aging signature of Svendsen[54] (aging signature 1) and the aging signature hub (https://agingsignature.webhosting.rug.nl/agingsignature) (aging signature 2) were enriched in middle-aged and aged WT in comparison to middle-aged and aged *Mda5*[-/-] HSCs (Fig. 4B–D). Ingenuity pathway analysis[55] of genes deregulated between aged WT and *Mda5*[-/-] HSCs showed enrichment for "eukaryotic translation elongation" and "EIF2 signaling" (Fig. 4E). Of note, we observed that a significant number of deregulated genes (247 genes) were close to induced peaks upon aging (Fig. 4F), suggesting that they experience changes in chromatin accessibility. Both aging signatures were enriched in the middle-aged or aged WT populations when compared to young WT (Fig. 4G). Interestingly, these signatures were enriched in middle-aged *Mda5*[-/-] when compared not only with young, but surprisingly also with aged *Mda5*[-/-] HSCs (Fig. 4H, I). Collectively, this comprehensive molecular analysis supports our functional analysis and shows that aged *Mda5*[-/-] HSCs are transcriptionally younger than their aged WT counterpart.

The master regulator of heat shock response and proteostasis is the transcription factor HSF1, which has been found to increase its concentration in the nucleus of aged and middle-aged HSCs[33]. We performed GSEA analysis using the transcriptomic signature from the study by Kruta et al.[33] (HSF1 signature), and we observed enrichment in middle-aged WT HSCs when compared to young WT or middle-aged *Mda5*[-/-] HSCs (Fig. 4J). Additionally, upstream regulator analysis from Ingenuity pathway analysis revealed HSF1 as an upstream regulator in all ages in WT and *Mda5*[-/-] HSCs (Fig. 4K), suggesting that it is an important regulator of HSC aging. We then performed Ingenuity pathway analysis for all the different comparisons. The categories for "eukaryotic translation elongation" or "EIF2 signaling" were enriched especially in *Mda5*[-/-] transcriptomes and at advanced ages. Finally, we observed enrichment of PKR signaling in middle-aged *Mda5*[-/-] HSCs (Supplementary Fig. 5A).

Thus, translation and cellular response to heat stress, both maintaining proteostasis, are important in *Mda5*[-/-] HSCs.

## Transposable elements are upregulated during aging

Since transposable elements (TE) are deregulated during aging[26], and we and others have previously shown that they can act as ligands for MDA5[19,20,22,23], we next analyzed our RNA-seq data for TE expression (Fig. 5A). We found that TE families were upregulated when comparing aged to young or to middle-aged HSCs. We observed 165 up and 3 downregulated TE families in aged WT HSCs compared to young HSCs, and 622 up and 2 down when comparing aged versus middle-aged WT HSCs (fold change 1.5, FDR < 0.05) (Supplementary Data 6). Interestingly, more TEs were upregulated in *Mda5*[-/-] HSCs. Indeed, 631 TE families were up, and 6 were downregulated in aged versus young *Mda5*[-/-] HSCs, and 764 TE families were up and 2 downregulated when we compared aged to middle-aged *Mda5*[-/-] HSCs. Very few or no differentially expressed TEs were detected between young (0),

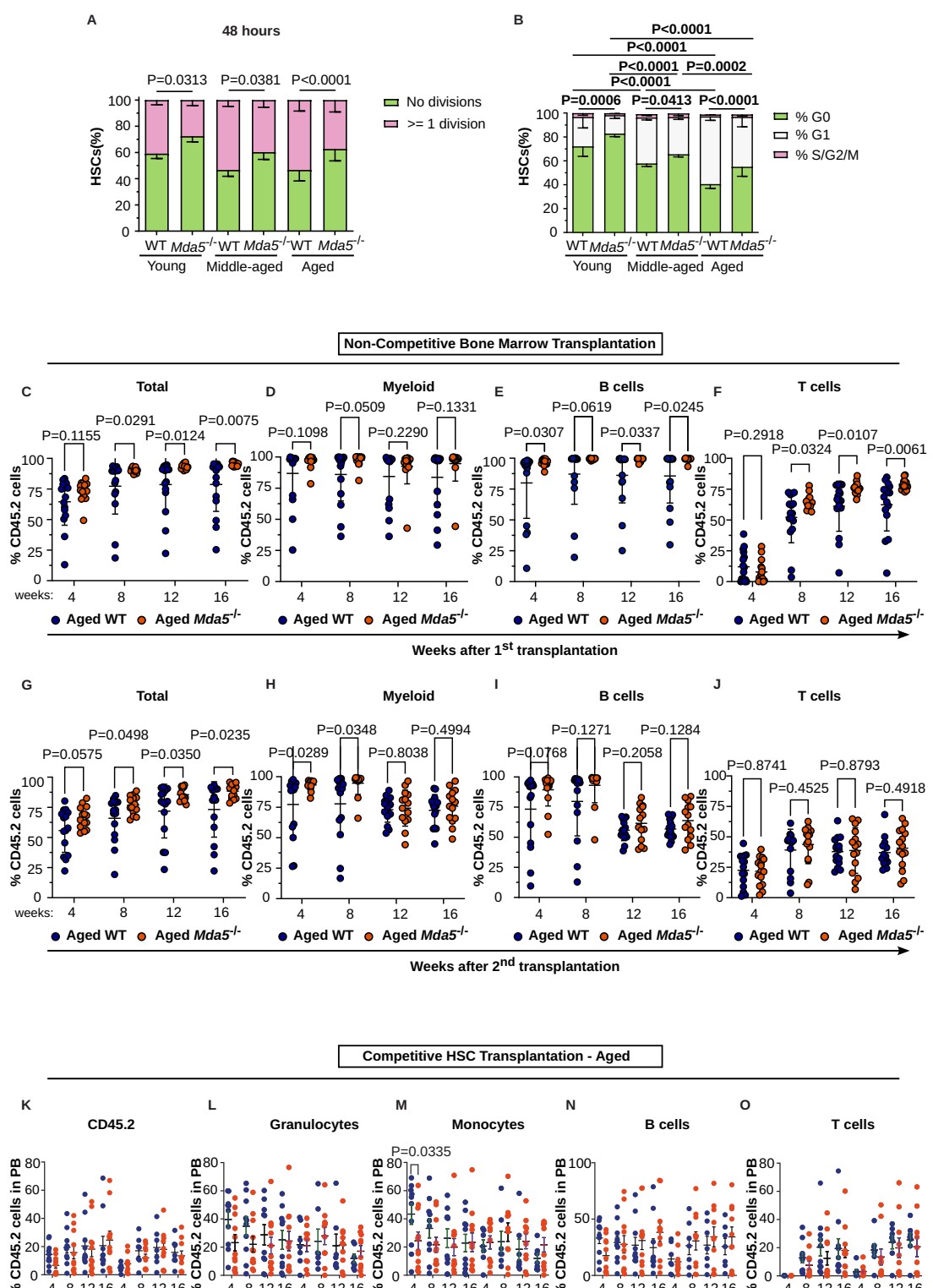

middle-aged (0), or aged (1) WT and *Mda5*[-/-] HSCs (Supplementary Data 6). Lastly, to verify the presence of double-stranded RNA in the cytoplasm of young, middle-aged, and aged WT and *Mda5*[-/-] HSCs, we performed a staining with anti-double-stranded RNA antibody. Indeed, we observed an increase in mean fluorescence intensity (MFI) in both WT and *Mda5*[-/-] HSCs upon aging (Fig. 5B–D).

## Single-cell RNA sequencing reveals a link between *Mda5* and translation

To understand the cellular and molecular heterogeneity, we performed single-cell RNA sequencing (scRNA-seq) in LSK cells isolated from young and aged WT and *Mda5*[-/-] animals. We applied known gene signatures to assign the cells to HSCs and MPP1–5 populations in both

**Fig. 3 | Aged *Mda5*-/- HSCs are more quiescent. A** Percentage of young (*N* = 6 WT and 7 *Mda5*-/-), middle-aged (*N* = 6 WT and 7 *Mda5*-/-), and aged (*N* = 14 WT and 14 *Mda5*-/-), WT or *Mda5*-/- HSCs that had undergone at least one division or no divisions after 48 h. *n* = 3 independent experiments. Data are presented as mean values ± SD. Two-tailed Fisher's exact test. *P* (aged WT vs. aged *Mda5*-/-) = 0.000000000009. **B** Cell cycle status of young (*N* = 5 WT and 7 *Mda5*-/-), middle-aged (*N* = 4 WT and 8 *Mda5*-/-), and aged (*N* = 7 WT and 6 *Mda5*-/-), HSCs. *n* = 3 independent experiments. Two-way ANOVA. Data are presented as mean values ± SD. *P* (aged WT vs. aged *Mda5*-/-) = 0.0000002962; *P* (young WT vs. middle-aged WT) = 0.0000320493; *P* (middle-aged WT vs. aged WT) = 0.0000000532; *P* (young *Mda5*-/- vs. middle-aged *Mda5*-/-) = 0.0000000006; *P* (young WT vs. aged WT) = 0.0000000005; *P* (young *Mda5*-/- vs. aged *Mda5*-/-) = 0.0000000005. Long-term donor hematopoietic (**C**), myeloid (**D**), B (**E**), and T (**F**) cell chimerism in the peripheral blood of recipient mice from primary non-competitive transplantations. (*N* = 16 WT and 15 *Mda5*-/-), biologically independent samples in *n* = 3 independent experiments. Each dot represents one mouse. Data are presented as mean values ± SD. Two-tailed unpaired t-tests. Long-term donor hematopoietic (**G**), myeloid (**H**), B (**I**), and T (**J**) cell chimerism in the peripheral blood of recipient mice from secondary non-competitive transplantations. (*N* = 15 WT and 14 *Mda5*-/-), biologically independent samples in *n* = 3 independent experiments. Each dot represents one mouse. Data are presented as mean values ± SD. Two-tailed unpaired t-tests. Competitive HSC transplant of aged WT and *Mda5*-/- HSCs. Long-term donor hematopoietic (**K**), granulocytic (**L**), monocytic (**M**), B-cell (**N**), and T-cell (**O**) chimerism in the peripheral blood of recipient mice from primary-HSC and secondary-bone marrow competitive transplantations. *N* = 14 for aged WT primary transplantation and *N* = 7 for secondary aged WT transplantation. *N* = 12 for aged *Mda5*-/- primary transplantation and *N* = 9 for secondary aged *Mda5*-/- transplantation. Two-tailed t-tests mean ± SD.

young and aged WT and *Mda5*-/- datasets[56–58] (Fig. 6A, B, and Supplementary Fig. 6A, B). Cells were then ordered by slingshot[59] pseudotime. The HSC population was placed at the top of the pseudotime ordering, followed by the MPP5/MPP1 or MPP5 populations, in agreement with previous findings that placed these populations in close proximity to HSCs both molecularly and biologically[56] (Fig. 6C, D and Supplementary Fig. 6C, D). Additionally, we performed pseudobulk analysis on the HSC/MPP1 cluster on young and aged *Mda5*-/- versus WT datasets, detecting 214 upregulated and 21 downregulated in the young and 57 upregulated and 25 downregulated in the aged HSCs (Fig. 6E and Supplementary Data 7, cut-offs: Log2FC 0.5 and FDR < 0.05). Concomitantly, analysis of aged WT and *Mda5*-/- versus young HSC/MPP1 populations revealed numerous deregulated genes (Fig. 6E and Supplementary Data 7).

To assess whether we could observe similar results in the multipotent progenitor populations, we clustered the MPP3 and MPP4 top 500 genes (Supplementary Fig. 6E, F and Supplementary Data 8–12). We observed 23 upregulated genes in aged *Mda5*-/- versus WT, while 276 were up and 49 were downregulated when comparing young *Mda5*-/- and WT MPP3 populations. Similarly, 20 genes were up, and 12 were downregulated in aged *Mda5*-/- versus WT, and 162 were up, and 66 were downregulated in young *Mda5*-/- and WT MPP4 populations (Supplementary Data 8–12). Ingenuity pathway analysis revealed an enrichment of "EIF2 signaling", "PKR-mediated signaling", and "unfolded protein response" categories in both HSC/MMP1 and MPP3, but not MPP4, aged populations. Additionally, "aggrephagy" and "cell cycle checkpoints" were enriched in young HSC/MPP1, MPP3, and MPP4 populations. Lastly, HSF1 was identified as an upstream regulator of deregulated genes in young HSC/MPP1, MPP3, and MPP4 and aged MPP3 (Fig. 6F).

Taken together, these results are in line with our bulk RNA-seq analysis and further show that the MPP3 population may undergo transcriptional changes similar to the HSC population, especially in terms of proteostasis and protein translation.

## Aged *Mda5*-/- HSCs exhibit enhanced PERK signaling

We have previously observed that EIF2 signaling is transcriptionally enriched in *Mda5*-/- HSCs. Stress in the endoplasmic reticulum (ER) leads to the unfolded protein response (UPR). The PERK-induced branch of the UPR is responsible for the phosphorylation of the EIF2a, which is in turn is responsible for blocking translation[60]. To delineate the order of events in our datasets, we performed GSEA analysis and observed that the endoplasmic-reticulum-associated protein degradation (ERAD pathway) is enriched both in the middle-aged and aged WT and *Mda5*-/- HSCs (Fig. 7A, B). However, the PERK branch was enriched only in the middle-aged and aged *Mda5*-/- HSCs (Fig. 7C). To assess EIF2a phosphorylation status downstream of the PERK pathway, we performed phospho-EIF2a staining in young, middle-aged aged and aged WT and *Mda5*-/- HSCs. Interestingly, phospho-EIF2a levels were comparable in young and middle-aged WT and *Mda5*-/- HSCs, but we detected an increased signal in aged *Mda5*-/- HSCs, verifying that this signaling is activated primarily in *Mda5*-/- HSCs (Fig. 7D).

## Aged *Mda5*-/- HSCs exhibit enhanced proteostasis

Since HSF1 and EIF2a are involved in the maintenance of protein homeostasis, we next assessed proteostasis in aged *Mda5*-/- and WT HSCs. We exploited two previously established assays to evaluate the abundance of misfolded and unfolded proteins, by quantifying ubiquitinated proteins and tetraphenylethene maleimide (TPE-MI)[39,61] labeled proteins by flow cytometry, respectively. Aged *Mda5*-/- HSCs exhibited a significantly lower amount of misfolded and unfolded proteins compared to the WT counterparts (Fig. 8A, B). We detected a similar trend in middle-aged *Mda5*-/- HSCs compared to WT (Supplementary Fig. 7A, B). The protein levels in young and aged WT and *Mda5*-/- HSCs were similar, albeit a bit more in the *Mda5*-/- HSCs (Supplementary Fig. 7C). To investigate whether the enhanced quiescence of *Mda5*-/- HSCs leads to fewer misfolded proteins, we combined the misfolded protein assay with the cell cycle staining. Interestingly, we observed reduced accumulation of misfolded proteins in G0 aged *Mda5*-/- HSCs compared to the WT counterpart (Supplementary Fig. 7D). Additionally, protein synthesis rate, measured using O-propargyl-puromycin (OP-Puro) incorporation, was significantly lower in aged *Mda5*-/- HSCs compared to aged WT HSCs (Fig. 8C). We hypothesized that these translational changes might have an impact on intracellular amino acid availability. Indeed, our intracellular metabolomic analysis on aged WT and *Mda5*-/- HSCs showed an increased concentration of amino acids in *Mda5*-/- HSCs (Fig. 8D). Collectively, these results align with the increased phospho-EIF2a observed in aged *Mda5*-/- HSCs (Fig. 7D) and support the hypothesis that amino acid accumulation in *Mda5*-/- HSCs is due to a reduced protein synthesis rate.

Autophagy has been implicated in the maintenance of protein homeostasis in response to stress[62,63]. Additionally, PERK signaling has been shown to lead to an induction of autophagy[35]. Recently, it has been shown that quiescent HSCs have a high autophagic flux, while aging has also been associated with impaired autophagy[37,39,64]. Consistent with these results, we observed a significant increase in the autophagic flux in aged *Mda5*-/- HSCs compared to WT (Fig. 8E). Taken together, these data indicate that aged *Mda5*-/- HSCs retain enhanced proteostasis and autophagic activity.

As a master regulator of protein homeostasis, HSF1 is activated under stress conditions and translocates into the nucleus to trigger a transcriptional response[30,31]. To evaluate HSF1 localization and abundance, we performed immunofluorescence staining on isolated young, middle-aged, and aged WT and *Mda5*-/- HSCs. HSF1 levels were similar in young WT and *Mda5*-/- HSCs, increased in middle-aged WT but not *Mda5*-/- HSCs, and significantly increased in aged *Mda5*-/- HSCs compared to age-matched WT counterparts (Fig. 8F, G). As we did not observe changes in HSF1 transcript levels upon aging (Supplementary Data 5), our results agree with Kruta et al.[33], in which they described

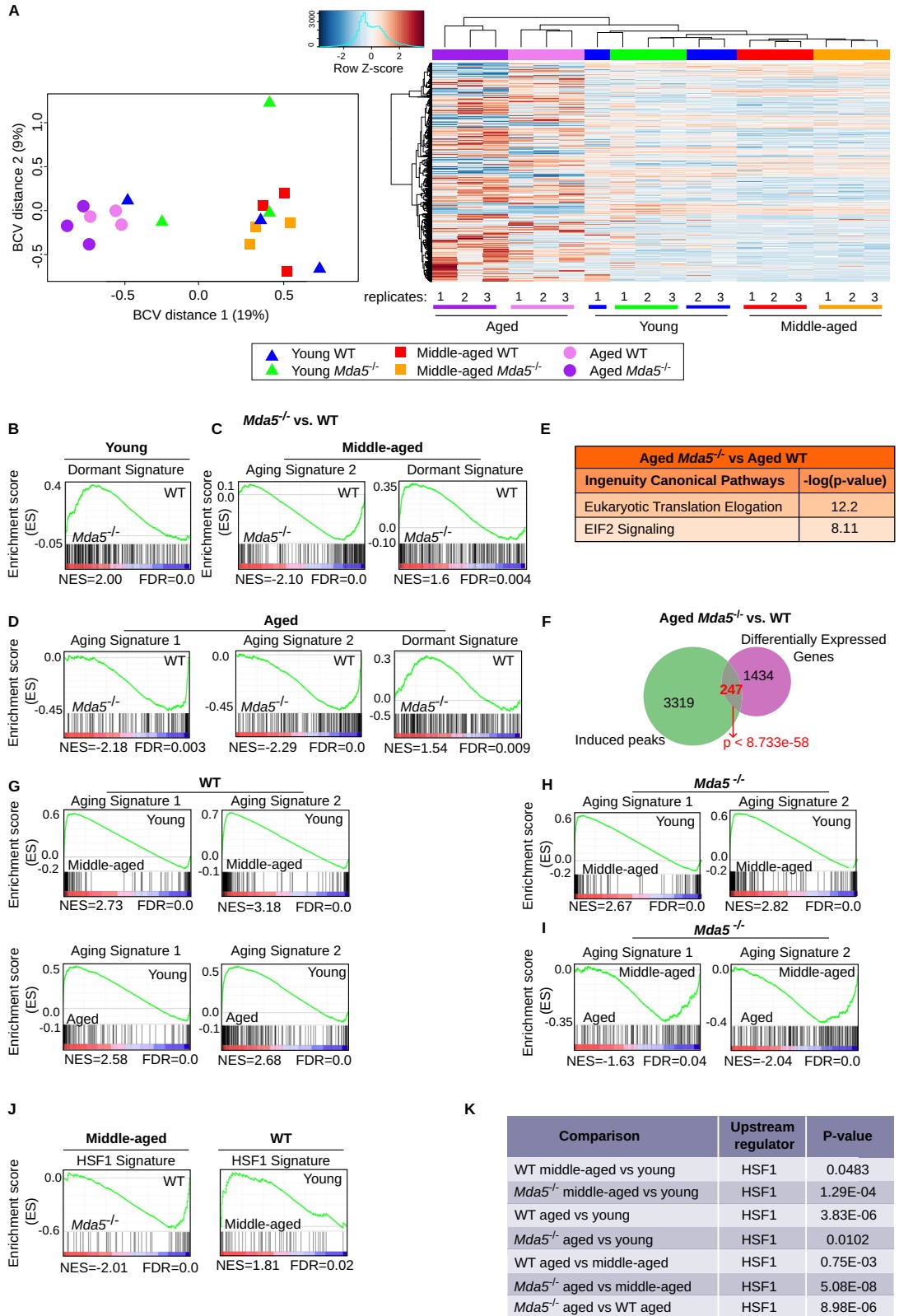

**Fig. 4 | HSF1 and EIF2 signaling are deregulated in *Mda5*-/- HSCs. A** Principal component analysis of expressed genes from young, middle-aged, and aged WT and *Mda5*-/- HSCs (BCV: biological coefficient of variation) (left). Heatmap of young, middle-aged, and aged WT versus *Mda5*-/- HSC top 500 most variably expressed genes, represented as z-scores from log-normalized counts per million (logCPM). (2500 cells per replicate, *n* = 3 replicates per sample-right). Gene set enrichment analysis (GSEA) of young (**B**), middle-aged (**C**), and aged (**D**) *Mda5*-/- (left) versus the respective WT (right) RNA-seq data using the indicated gene signatures. **E** Ingenuity pathway analysis of canonical pathways of aged *Mda5*-/- versus WT HSCs. **F** Venn diagram depicting the overlap between the deregulated genes in aged WT versus *Mda5*-/- HSC and the induced peaks in aged *Mda5*-/- HSC ATACseq. The hypergeometric test shows the significance of the overlap. **G**–**J** Gene set enrichment analysis (GSEA) of the indicated transcriptomes using the indicated gene signatures. **K** Ingenuity pathway upstream regulator analysis for HSF1 at the indicated transcriptomes.

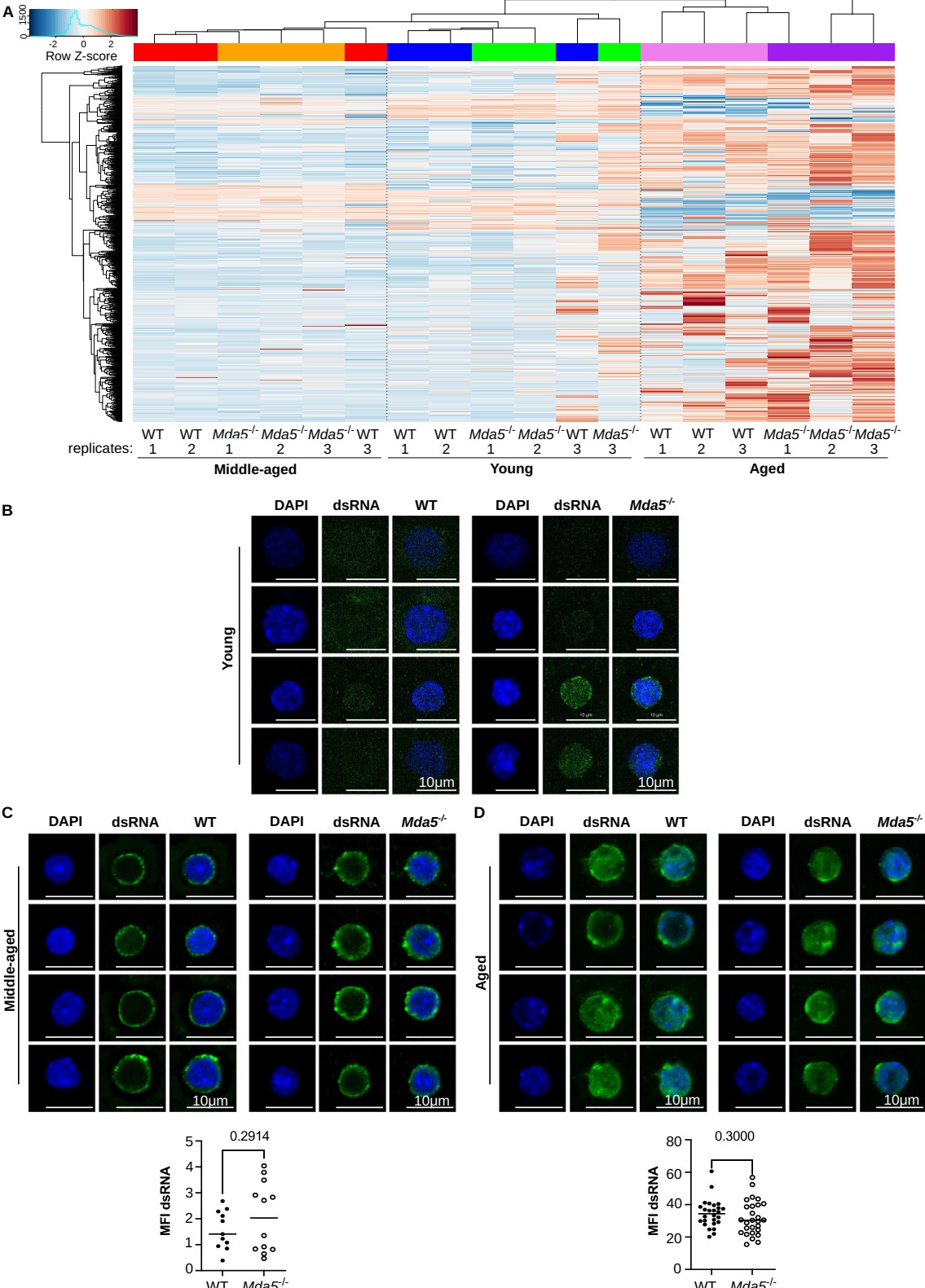

**Fig. 5 | Transposable elements are increased during aging. A** Heatmap depicting the z-score of expression of transposable elements from bulk RNA seq for young, middle-aged, and aged *Mda5⁻ᐟ⁻* and WT HSCs. Double-stranded RNA staining in young (**B**), middle-aged (**C**), and aged (**D**) HSCs (scale bar 10 μm). Quantification of mean fluorescence intensity (MFI) of dsRNA in HSCs in middle age **C** ($N = 11$ WT and 12 *Mda5⁻ᐟ⁻*HSCs) and aged **D** ($N = 26$ WT and 26 *Mda5⁻ᐟ⁻* HSCs) WT or *Mda5⁻ᐟ⁻* HSCs, $n = 2$ and $n = 3$ independent experiments respectively, each dot represents one cell. Two-tailed t-tests, median.

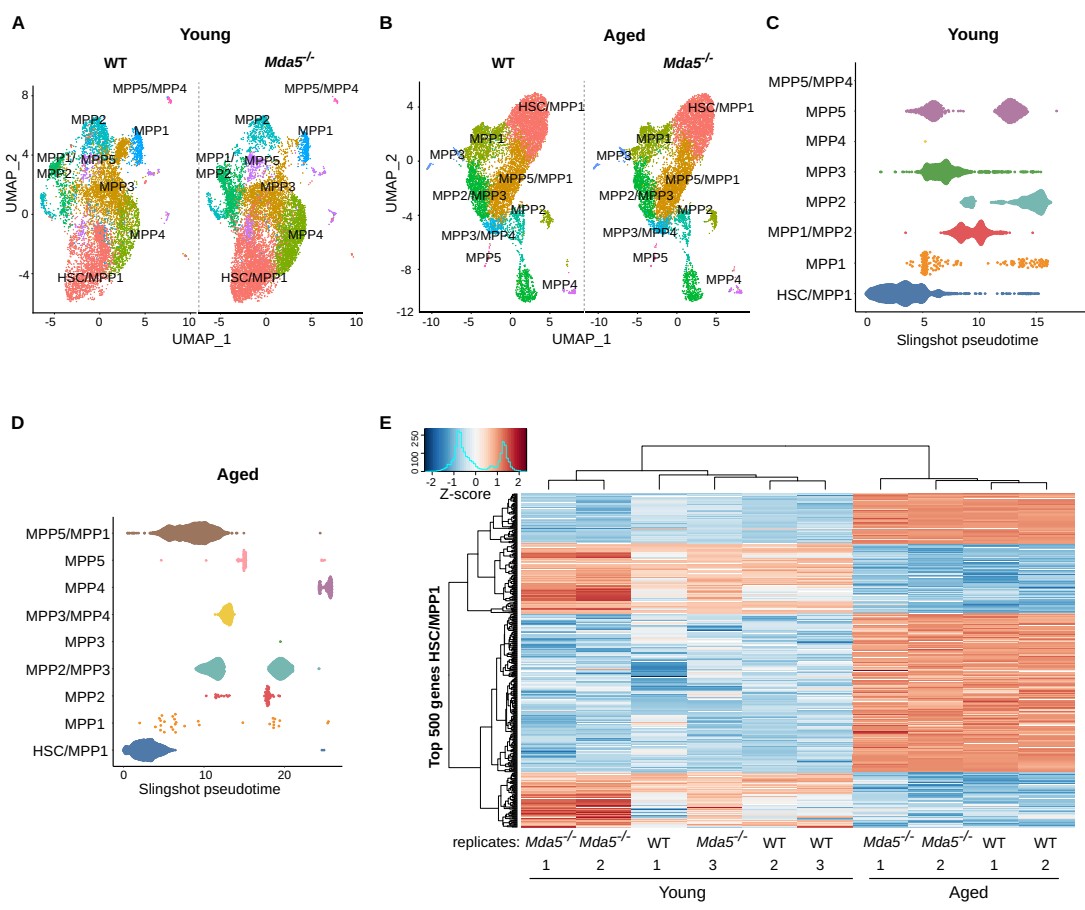

**Fig. 6 | scRNA-seq of LSK WT and *Mda5*⁻/⁻ cells.** Uniform manifold approximation and projection (UMAP) projections of WT and *Mda5*⁻/⁻ LSK single-cell transcriptomes from three young (**A**) and two aged (**B**) donors, respectively, highlighting the different hematopoietic populations. Differentiation trajectory of cell type clusters represented in pseudotime ordering inferred by slingshot for integrated young (**C**) and aged (**D**) datasets. **E** Heatmap top 500 most variably expressed genes, represented as z-scores from log-normalized counts per million (logCPM) detected by pseudobulk analysis on the HSC/MPP1 cluster for young and aged *Mda5*⁻/⁻ and WT HSCs. **F** Ingenuity pathway analysis (IPA) of canonical pathways for the indicated comparisons.

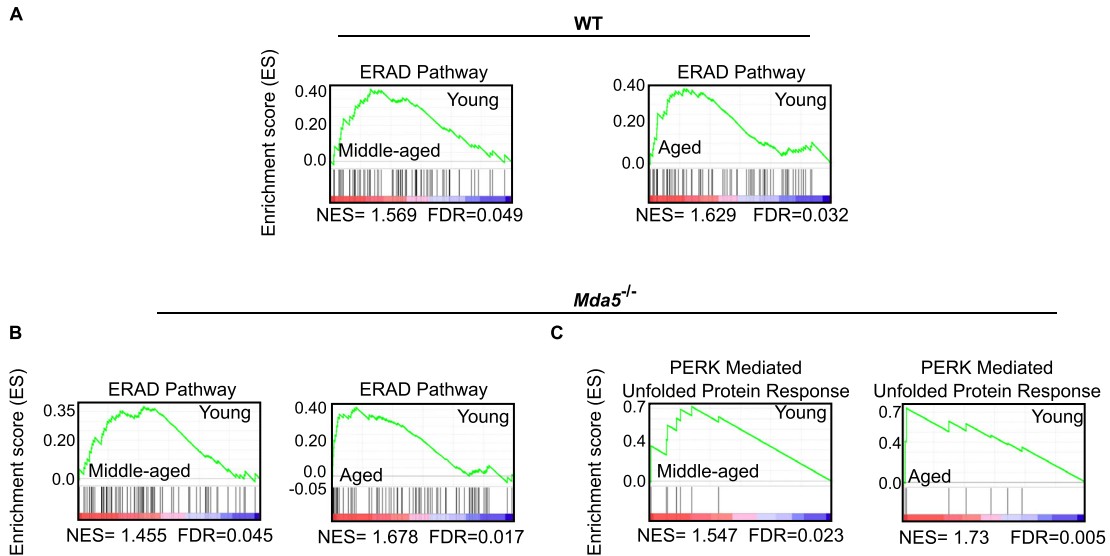

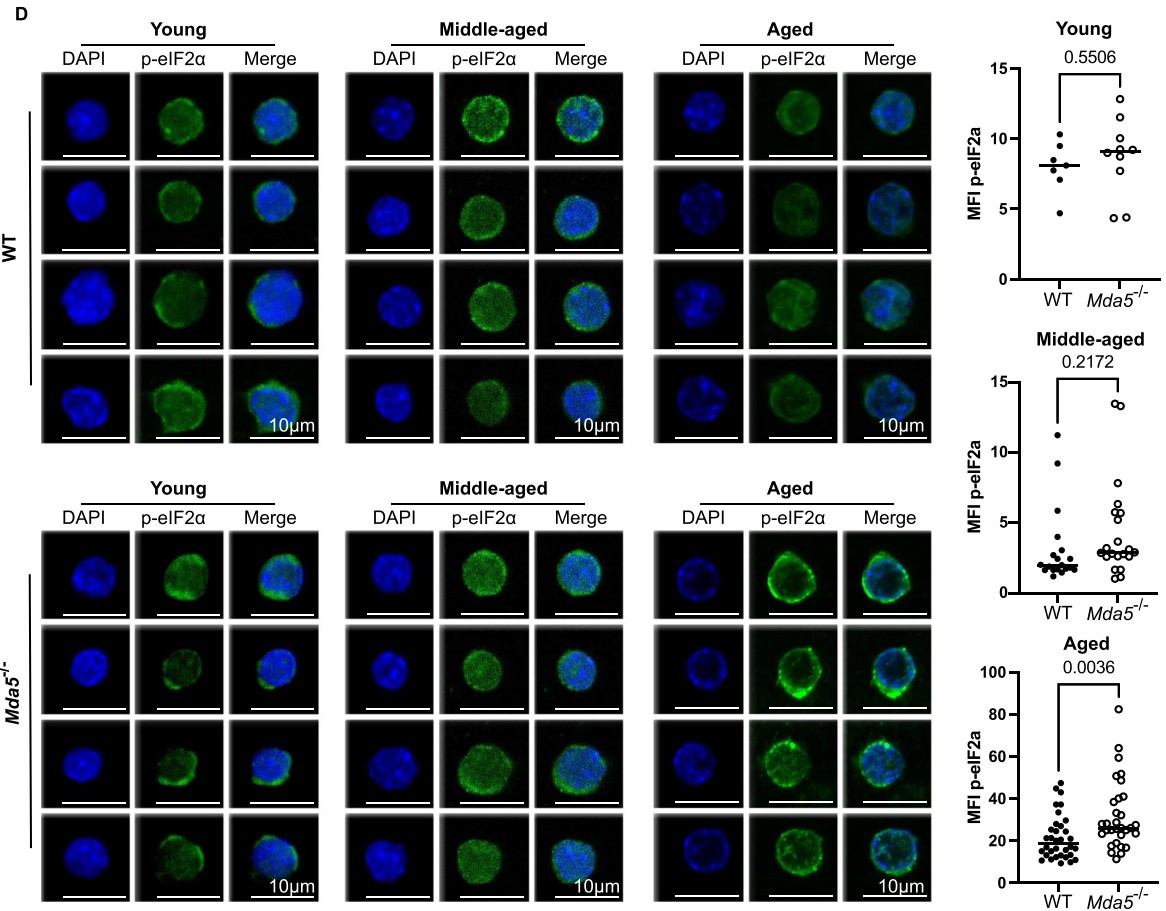

**Fig. 7 | EIF2 is deregulated in *Mda5⁻/⁻* HSCs.** Gene set enrichment analysis (GSEA) of WT (**A**) middle-aged versus young and aged versus young and *Mda5⁻/⁻* (**B**, **C**) middle-aged versus young and aged versus young RNA-seq data using the indicated gene signatures. **D** Phospho-EIF2a staining in young, middle-aged, and aged WT and *Mda5⁻/⁻* HSCs (scale bar 10 μm). Quantification of mean fluorescence intensity (MFI) of phospho-EIF2a in HSCs in young (*N* = 7 WT and 10 *Mda5⁻/⁻* HSCs in *n* = 1 independent experiment), middle-aged (*N* = 19 WT and 21 *Mda5⁻/⁻* HSCs in *n* = 3 independent experiment), and aged WT or *Mda5⁻/⁻* HSCs, (*N* = 34 WT and 31 *Mda5⁻/⁻* HSCs in *n* = 3 independent experiment), each dot represents one cell. Two-tailed t-tests, median.

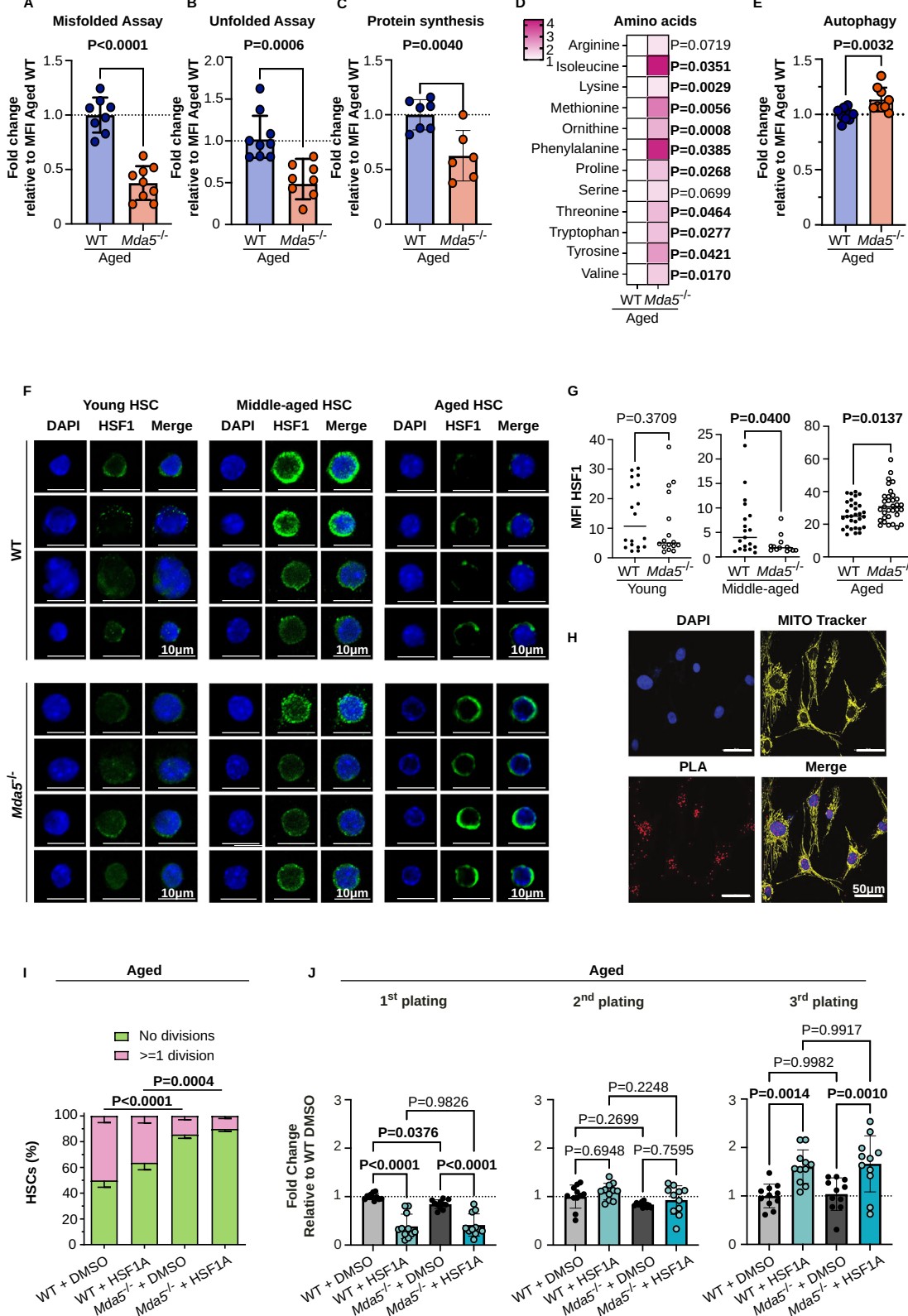

HSF1 upregulation, at least at the protein level, upon aging and already in middle-aged cells. To further strengthen these results, we performed a similar staining using an antibody against phosphorylated HSF1 (pSer326), which marks HSF1 activation. Similarly, we observed reduced accumulation of phospho-HSF1 in middle-aged *Mda5*[-/-] HSCs. Increased levels of phospho-HSF1 were detected in aged *Mda5*[-/-] HSCs, albeit this result was not significant (Supplementary Fig. 6E, F).

To investigate whether MDA5 can retain HSF1 in the cytoplasm, we overexpressed MDA5 in HEK293T cells and stained for HSF1 to assess its localization. In HEK293T cells, HSF1 is localized in the nucleus. We observed that overexpression of full-length MDA5 was enough to keep a portion of HSF1 in the cytoplasm (Supplementary Fig. 7G, H). However, HSF1 was not tethered in the cytoplasm upon overexpression of a mutant that lacks both CARD domains

**Fig. 8 | Aged *Mda5*[-/-] HSCs exhibit enhanced proteostasis.** Relative accumulation of misfolded (**A** N = 8 WT and 9 aged *Mda5*[-/-] HSCs in n = 3 independent experiment) and unfolded (**B** N = 9 WT and 8 aged *Mda5*[-/-] HSCs in n = 3 independent experiment) proteins in aged WT or *Mda5*[-/-], normalized to the corresponding WT mean fluorescence intensity (MFI). Each dot represents one mouse. Data are presented as mean values ± SD. Two-tailed unpaired t-tests. For **A**: P (aged WT vs. aged *Mda5*[-/-]) = 0.0000007. **C** Relative protein synthesis in aged WT or *Mda5*[-/-], normalized to the corresponding WT mean fluorescence intensity (MFI). N = 7 WT and 6 aged *Mda5*[-/-] HSCs biologically independent samples in n = 2 independent experiments. Each dot represents one mouse. Data are presented as mean values ± SD. Two-tailed unpaired t-tests. **D** Fold change of intracellular amino acid concentrations of aged WT or *Mda5*[-/-] HSCs, relative to WT control. N = 4 WT and 4 aged *Mda5*[-/-] for arginine, isoleucine, N = 7 WT and 6 aged *Mda5*[-/-] for lysine, N = 4 WT and 5 aged *Mda5*[-/-] for methionine and ornithine, N = 5 WT and 6 aged *Mda5*[-/-] for phenylalanine, N = 5 WT and 5 aged *Mda5*[-/-] for proline, N = 4 WT and 7 aged *Mda5*[-/-] for serine, N = 5 WT and 6 aged *Mda5*[-/-] for threonine, N = 3 WT and 7 aged *Mda5*[-/-] for tryptophan, N = 3 WT and 3 aged *Mda5*[-/-] for tyrosine, and N = 6 WT and 6 aged *Mda5*[-/-] for valine in n = 3 independent experiments. Data are presented as mean values ± SD. Two-tailed unpaired t-tests. **E** Relative autophagic flux in aged WT or *Mda5*[-/-], normalized to the corresponding WT mean fluorescence intensity (MFI). N = 10 WT and N = 9 *Mda5*[-/-] biologically independent samples in n = 3 independent experiments. Each dot represents one mouse. Data are presented as mean values ± SD. Two-tailed unpaired t-tests. **F** Representative immunofluorescence staining showing HSF1 expression in young, middle-aged, and aged WT or *Mda5*[-/-] HSCs (Scale bar 10 μm). **G** Quantification of mean fluorescence intensity (MFI) of HSF1 in HSCs in young (N = 14 WT and 21 *Mda5*[-/-] HSCs), middle-aged (N = 19 WT and 12 *Mda5*[-/-] HSCs), and aged (N = 31 WT and 33 *Mda5*[-/-] HSCs), n = 3 independent experiments, each dot represents one cell. Two-tailed t-tests, median. **H** Proximity ligation assay for MDA5 and HSF1. PLA spots can be detected in red. The nucleus is stained with DAPI, and the cytoplasm with Mitotracker in yellow. Scale bar = 50 μm. Representative figures from one of the three experiments done. **I** Percentage of DMSO (N = 5 WT and 6 *Mda5*[-/-]) or HSF1A-treated (N = 5 WT and 5 *Mda5*[-/-]) aged WT or *Mda5*[-/-] HSCs that had undergone at least one division or no divisions after 48 h. n = 2 independent experiments. Data are presented as mean values ± SD. Two-tailed Fisher's exact test. P (aged WT DMSO vs. aged WT HSF1A) = 0.000007. **J** Fold change of colonies formed by either aged WT or *Mda5*[-/-] HSCs, cultured in Methocult treated with either DMSO or HSF1A. Normalized to DMSO-treated WT HSCs. N = 11 biologically independent samples in n = 4 independent experiments. Data are presented as mean values ± SD. Two-way ANOVA. P (aged WT DMSO vs. aged WT HSF1A) < 0.000000000000001; P (aged *Mda5*[-/-] DMSO vs. aged *Mda5*[-/-] HSF1A) = 0.000000000001341.

(Supplementary Fig. 7G, H). Lastly, proximity ligation assay revealed that MDA5 and HSF1 are found in proximity (Fig. 8H), suggesting a potential interaction between these two proteins. To our surprise, we observed both nuclear and cytoplasmic staining for the proximity ligation assay, hinting at the fact that MDA5 may also be in the nucleus.

Because HSF1 protein levels were increased and localized in the nucleus in *Mda5*[-/-] HSCs, we reasoned that HSF1 activation could lead to the alteration of *Mda5*[-/-] and WT phenotypes, respectively. HSF1A is a small molecule inhibitor of TRiC chaperonin (chaperonin that keeps HSF1 in the cytoplasm) and functions as HSF1 activator (HSF1A), by preventing TRiC binding to HSF1, thus permitting its trimerization and translocation to the nucleus[33,65]. Therefore, we reasoned that the HSF1A inhibitor could induce aged WT HSCs to behave like aged *Mda5*[-/-] HSCs. Indeed, HSF1A treatment enhanced aged WT HSC quiescence by decreasing their ability to divide in culture (Fig. 8I). In addition, we performed serial colony-forming unit (CFU) assays by plating HSCs in methylcellulose and replating a similar number of cells every week. This assay evaluates the in vitro repopulation capacity of HSCs. Indeed, the inhibitor of HSF1 that made the cells more quiescent led to the enhanced capacity of aged WT HSCs to produce colonies after the third replating, suggesting that they can keep their in vitro self-renewal capacity (Fig. 8J).

These data suggest that a lack of *Mda5* can delay the aging process in the HSC compartment by retaining effective proteostasis through HSF1 engagement and reduction of protein translation.

## Discussion

Increased inflammatory signaling is a hallmark of aging, and manipulation of inflammaging has been proposed as a strategy to alleviate symptoms of aging[66]. DNA sensors like cGAS (GMP-AMP (cGAMP) synthase) and the cGAS/STING (stimulator of interferon genes) pathway have been implicated in age-related senescence[67] and inflammation, especially in the brain in conjunction with neurodegeneration[68-70]. However, the role of RNA sensors and the causal relation between inflammaging and tissue dysfunction in aging remains largely unknown. Here, we examined the function of the innate immune sensor MDA5 in delaying hematopoietic aging. Since TEs and mitochondrial RNA can activate MDA5[19,20,22,23,71,72], it is therefore possible that the accumulation of TEs observed during aging by this and other studies[26,73] or mitochondrial RNA, leads to activation of MDA5 signaling. This activation is evidenced by the decrease in multiple inflammatory cytokines in the absence of MDA5. However, the function of inflammation in blood rejuvenation is currently

understudied. Parabiosis experiments have implicated inflammatory signals indirectly since rejuvenation of multiple tissues can be achieved with the transfusion of young blood[74], while aging features can be induced in young animals by old blood[75-78]. Other studies showed that aged HSCs are refractory to systemic rejuvenation intervention delivered via the blood stream[79]. In another study, the microbiome/IL-1/IL-1R1 axis was proven to be key for hematopoietic aging, and manipulation of this axis could lead to better repopulation capacity and reversal of myeloid bias[7]. However, constant treatment with PolyI:C, which activates MDA5, has been shown to lead to hematopoietic aging[15]. Here, we provide evidence that a lack of MDA5 delays HSC aging, enhances the repopulation capacity, and counteracts myeloid bias, but also improves various metabolic aspects of HSCs isolated from aged mice. Although we primarily focused on the intrinsic properties of HSCs, we anticipate that low-grade inflammation will lead to the amelioration of aging phenotypes in other cell populations in the bone marrow and the bone marrow niche. Collectively, this could contribute to the retention of a youthful phenotype as seen in *Mda5*[-/-] animals.

HSC metabolism has key differences between development and aging[80,81]. Decreased autophagy[37] and proteasomal degradation[82], enhanced mitochondrial oxidative phosphorylation and increased ROS production[83,84] are characteristics of younger HSCs. Additionally, mitochondrial potentiation ameliorates age-related heterogeneity in HSC function[48]. Here we see increased mitochondrial mass and TMRM, with additional increased autophagy and reduced translation. Specific metabolites that have been associated with youthful phenotypes, like decreased GSSG[43] and increased NAD+ and NADP+[44], attest to a younger phenotype of HSPCs. Amino acids that are usually in excess in MPPs rather than in HSCs[85] are elevated in *Mda5*[-/-] HSCs that also show low translational output, usually attributed to HSCs rather than MPPs[28,86].

Overall, transcriptional changes observed in aged *Mda5*[-/-] HSCs versus WT pointed to HSF1 as an upstream regulator and the UPR being deregulated in *Mda5*[-/-] HSCs. Indeed, HSF1 has been instrumental in HSCs upon stress or aging[33]. Here we show that HSF1 activation can induce quiescence and improve in vitro repopulation capacity in WT HSCs, in line with the in vivo results described by Kruta et al.[33]. Proximity ligation assay showed that MDA5 is proximal to HSF1, and chaperones of HSF1, like HSP40, have been previously found to interact with MDA5[87]. It is possible that these interactions keep HSF1 away from chromatin. This is also in agreement with our experiment, in which we observed partial retention of HSF1 in the cytoplasm upon

MDA5 overexpression. An alternative scenario could be that HSF1 is activated in middle-aged animals, as previously shown[33], and is retained in *Mda5*[-/-] HSCs due to their quiescent phenotype and the consequent decreased divisional history. However, in this case, any manipulation that retains HSCs in quiescence should lead to increased proteostasis. Indeed, very few data exist on the transcriptional signature of aged HSCs with deficiency of inflammatory factors that usually remain more quiescent than their WT counterparts. The transcriptional signature of aged IL1R1-KO HSCs does not contain any proteostatic terms[7]. These results do not preclude that a non-cell-autonomous mechanism exists, and it would be interesting to examine whether youthfulness of other tissues besides the hematopoietic system happens in the absence of MDA5. Additionally, our results show that proteostasis is maintained by phosphorylation of EIF2a in the aged *Mda5*[-/-] HSCs. This could be due to the higher activation of PERK signaling, as we see in our transcriptomic signatures, but it could also be due to the activation of PKR by TEs, since we see significantly more activation of PKR in middle-aged *Mda5*[-/-] HSCs.

A caveat of our manuscript is the inability to say if the function of MDA5 in aging starts during aging or much earlier, when the organism is young. Our experiments point to many changes in middle-aged and aged *Mda5*[-/-] HSCs that should contribute to a scenario where the function of MDA5, at least in proteostasis, starts at middle age. However, the transplantation experiments argue for better function of *Mda5*[-/-] HSCs from a young age.

Mutations in IFIH1/MDA5 in humans are associated with Singleton-Merten Syndrome and Aicardi–Goutières syndrome. Moreover, IFIH1 SNPs (single-nucleotide polymorphisms) are associated with an increased risk of type 1 diabetes. Antibodies against MDA5 are associated with amyopathic dermatomyositis with rapidly progressive interstitial lung disease. Most of the diseases were associated with mutations that enhance the function of MDA5. Here, we propose that the absence of MDA5 leads to delayed aging. A recent preprint describes the whole-body delayed aging of *Mda5*[-/-] animals and shows improved motor functions and longer lifespan in aged *Mda5*[-/-] mice. Indeed, in this study, the frailty test showed that *Mda5*[-/-] mice are less frail and perform better in a grip test. They also have a longer lifespan[88]. It is difficult to mimic an aging study in human cells. As previously stated, *Mda5* is not induced during aging in either mouse or human. However, we could envision a situation in which MDA5 absence, low inflammation, induced proteostasis, low translation, and higher autophagy would be beneficial for human health and life span.

Altogether, our data point to an interplay between the innate immune sensing pathways and protein homeostasis mechanisms in HSCs. This is common in long-lived animals like bats, which exhibit low inflammation, higher autophagy, and increased expression of heat shock proteins to maintain proteostasis[89]. In agreement with these observations, we show that the absence of MDA5 leads to delayed aging, and this is at least partly due to retained proteostasis.

# Method
## Mice
All mouse experiments were carried out in accordance with the guidelines of the Federation of European Laboratory Animal Science Association and following legal approval of the Regierungspräsidium Freiburg and the Animal Care and Use Committee of the French authorities-Ciepal Azur (35-9185.81/G-18/127, 35-9185.81/G-22/058, PEA845 or apafis 2022062916507603_v5). All of the animals were maintained at the animal facility of the Max Planck Institute of Immunobiology and Epigenetics or at the IRCAN facility under specific pathogen-free conditions in individually ventilated cages with a light-dark cycle of 12–12 h at 20–24 °C under 45–65% humidity. For all genotypes, gender matched female or male mice were used in the experiments, and experimental and control mice were co-housed in

the same facility rooms. Mice were euthanized with cervical dislocation or CO$_2$ followed by cervical dislocation. HSC transplantations were performed at St. Jude Children's Hospital. The transplantation procedure was performed according to protocols approved by the St. Jude Children's Research Hospital Institutional Animal Care and Use Committee, under animal protocol #3096.

*Mda5*[-/-] mice or B6.129X1(C)-*Ifih1*[tm1.1Cln]/J were purchased from Jackson Laboratory Strain #:015812. CD45.1 BL6 mice were bred in MPI-IE unless otherwise stated (Jax Strain #:002014). CD45.1 JAXBoy mice were purchased from Jackson Laboratory (C57BL/6J-*Ptprc*[em6Lutzy]/J, JAX stock #033076).

## Antibodies
The following antibodies were purchased from BioLegend and used at a dilution 1:500 dilution unless stated otherwise: anti-CD45.2/Ly5.2 (Pacific Blue or FITC, 104); anti-CD45.1/Ly5.1 (Alexa Fluor 700 or Pe/Cy7, A20); anti-CD45 (FITC, 1:1000, 30-F11); anti-CD45R/B220 (BV650 or Alexa Fluor 700 or FITC or PE/Cy7 or biotin, RA3-6B2); anti-Ly6G/Ly6C (Gr1, BV650 or PE/Cy7 or APC, 1:1000, FITC or biotin, 1:1600, RB6-8C5); anti-CD11b (BV650 or PE/Cy7 or APC/Cy7, 1:1000, FITC or biotin, 1:1600, M1/70); anti-TER119 (BV650 or PE/Cy7, 1:1000, FITC or biotin, 1:1600, TER119); anti-CD3ε (BV650 or PE/Cy7, 1:1000, FITC or biotin, 1:1600, 145-2C11); anti-NK-1.1 (FITC, PK136); anti-CD19 (FITC, 1D3/CD19; anti-CD4 (PE/Cy5 or FITC, 1:1000, RM4-5); anti-CD8a (PE/Cy5 or FITC, 1:2000, 53-6.7); anti-CD117 (cKit, BV711, 1:1000, 2B8); anti-Ly-6A/E (Sca1, PE/Cy7 or APC/Cy7, 1:400, E13-161.7); anti-CD201 (EPCR, PE or APC, 1:200, RCR-16); anti-CD150 (SLAM, PE/Dazzle™ 594 or BV421, 1:400, TC15-12F12.2); anti-CD48 (PE/Cy7 or APC/Cy7, 1:400, or BV421, 1:1000, HM48-1); anti-CD34 (FITC, 1:50, SA376A4); anti-CD135/Flk2 (PE or APC, 1:200, A2F10); anti-Ki67 (FITC or PE, 1:100, 16A8); anti-CD16/32 (APC or PE, 1:1000, 93); anti-CD127 (IL-7Rα, APC or PE, 1:1000, A7R34); goat anti-mouse IgG, IgM (H + L) secondary (1:2500, Alexa Fluor 488, A-10680, Invitrogen) See also Supplementary Table 1.

## Sorting strategy
Throughout the text, HSCs refer to EPCR SLAM cells (Lin⁻EPCR⁺CD150⁺CD48⁻ unless otherwise stated. LSK SLAM: Lin⁻Sca1⁺cKit⁺(LSK) CD150⁺CD48⁻; LT-HSCs: LSKCD150⁺CD48⁻CD34⁻CD135⁻; MPP1: LSKCD150⁺CD48⁻CD34⁺CD135⁻; MPP2: LSKCD150⁺CD48⁺CD34⁺CD135⁻; MPP3: LSKCD150⁻CD48⁺CD34⁺CD135⁻; MPP4: LSKCD150⁻CD48⁺CD34⁺CD135⁺.

## HSC isolation, flow cytometry and cell sorting
Tibiae, femurs, and hip bones were isolated and crushed in PBS. Cells were washed with FACS buffer (PBS, 2% FCS, 1 mM EDTA) and counted Vi-cell XR counter (Beckman Coulter).

-For lineage quantification, HSPC characterization, $3 \times 10^6$ cells were stained with antibody mix in FACS buffer for 30 min at 4 °C. Sample acquisition was performed using a Fortessa FACS analyzer (BD Biosciences). All data were analyzed using FlowJo (BD) software.

-For sorting, samples were enriched by lineage depletion using a biotin-conjugated lineage antibodies cocktail (CD3ε, CD11b, CD45R/B220, Ly6G/Ly6C, TER-119) for 20 min at 4 °C. Streptavidin nanobeads (MojoSort, 480016, BioLegend) were added for 20 min at 4 °C, with subsequent magnetic separation for 5 min at room temperature. Enriched samples were stained with an antibody mix in FACS buffer against EPCR, CD150, and CD48 for 30 min at 4 °C. Cells were washed and resuspended in 1 ml FACS buffer, and sorting was performed on a BD FACSAriaIII or a BD FACSAriaFusion (BD Biosciences).

## Ex vivo HSC culture
HSCs or lineage-depleted cells were cultured in StemPro-34 SFM medium with 2.5% StemPro-34 Supplement (Gibco), 50 ng/ml mSCF, 25 ng/ml mTPO, 30 ng/ml mFlt3L, 1% penicillin-streptomycin, and 2 mM l-glutamine.

## Single-cell division assay

HSCs were single sorted on Terasaki microtest plates (654102, Greiner) in the medium described above. For HSF1 treatment, DMSO or HSF1A (8 μM final concentration, Axon 1890, Axon MedChem) was added to the medium before sorting. The frequencies presented in the single cell division assay plots were calculated for each Terasaki plate separately and independently as follows:

Frequency of HSCs that did not divide = number of wells with one sorted HSC that did not divide/total number of wells in the Terasaki plate with live sorted HSCs

Frequency of HSCs that divided at least once = number of wells with sorted HSC that did divide/total number of wells in the Terasaki plate with live sorted HSCs

Empty wells or wells containing dead cells were not included in the frequency calculations. Fisher's exact test was performed considering the total number of wells across all different independent experiments.

## Cell cycle staining

Samples were isolated and enriched as described above. $3 \times 10^6$ lineage-negative cells were stained with the antibody mix in FACS buffer for 30 min at 4 °C to identify HSCs. Then, cells were washed and resuspended in fixed intracellular Fixation Buffer (00-8222-49, ThermoFisher Scientific) for 10 min at 4 °C. Samples were then washed and resuspended in permeabilization buffer (00-8333-56, ThermoFisher Scientific) with anti-Ki67 antibodies for 2 h at 4 °C. Lastly, cells were washed and resuspended in PBS with Hoechst 33258 (H3569, Life Technologies) at room temperature for at least 15 min before acquisition. Sample acquisition was performed using a Fortessa FACS analyzer (BD Biosciences). All data were analyzed using FlowJo (BD) software.

## Colony-forming unit assay

Two hundred HSCs were sorted in 1 ml Mouse MethoCult GF medium (M3434, StemCell Technologies) and plated in 35 mm petri dishes. DMSO, HSF1A (8 μM final concentration, Axon 1890, Axon MedChem) was added to the medium before each plating. Colonies were counted after 5–7 days at 37 °C and 5% $CO_2$. For replating, cells were washed with PBS, counted, and replated (10,000 cells of each replicate) in MethoCult GF (M3434, StemCell Technologies). Colonies were quantified after 4–5 days, after the second and third replating.

## Whole bone marrow transplantations

Aged WT and *Mda5*$^{-/-}$ mice (CD45.2$^+$/Ly5.2) were used as donors, young WT mice (CD45.1$^+$/Ly5.1) were used as competitors in competitive transplantations and as recipients in non-competitive transplantations. Young WT mice (CD45.1$^+$/CD45.2$^+$) were used as recipients in competitive transplantations. We calculated the HSC frequency of the donors and intravenously injected the volume of bone marrow containing 300 HSCs into lethally irradiated (9.5 Gy) recipients. In competitive transplantations, we additionally injected 500,000 competitor-derived bone marrow cells. Peripheral blood chimerism was checked every 4 weeks after transplantation for 16 weeks. Briefly, 20 μl of blood was obtained from the submandibular vein using a lancet. Erythroid cells were lysed in ammonium-chloride-potassium (ACK, BP10-548E, Lonza Bioscience) for 5 min at room temperature. Cells were washed with PBS and stained for 30 min at 4 °C with anti-CD45.1/Ly5.1, anti-CD45.2/Ly5.2, anti-CD4, anti-CD8a, anti-CD11b, anti-Ly6G/Ly6C, anti-CD45R/B220, and anti-Ter119 antibodies. After washing and resuspending in FACS buffer, samples were acquired on a Fortessa FACS analyzer (BD Biosciences). All data were analyzed using FlowJo (BD) software.

## HSC transplantations

Transplantations were performed using the CD45.1/CD45.2 congenic system. Recipient CD45.1 JAXBoy mice (C57BL/6J-*Ptprc*$^{em6Lutzy}$/J, JAX

stock #033076) (10 weeks old) were lethally irradiated with 11 Gy delivered in a split dose of 5.5 Gy each, at least 3 h apart, using a gamma irradiator with a 137Cs source. Recipients were treated with Sulfamethoxazole/Trimethoprim in drinking water for 7 days before the transplant and for 21 days post-transplant. On the day of the transplant, 100 FACS-sorted donor HSCs (Lin⁻Sca1⁺c-kit⁺CD135⁻CD48⁻CD150⁺) were injected into the tail vein along with $2 \times 10^5$ of whole bone marrow cells isolated from age-matched CD45.1 JAXBoy mice. Hematopoietic reconstitution in peripheral blood was monitored by flow cytometry every 4 weeks, and the primary recipients were euthanized 16 weeks post-transplant. For the secondary transplantation, bone marrow of primary recipients injected with HSCs from the same donor mice was pooled, and $3 \times 10^6$ of pooled whole bone marrow cells were delivered via tail vein injections into lethally irradiated CD45.1 JAXBoy secondary recipients. Hematopoietic reconstitution in peripheral blood was monitored by flow cytometry every 4 weeks. The secondary recipients were euthanized 16 weeks post-transplant. The following cell populations were analyzed in the peripheral blood and bone marrow of transplant recipients using flow cytometry: granulocytes (Mac1⁺Gri^hi), monocytes (Mac1⁺Gri^lo), B cells (CD19⁺), T cells (CD3⁺), LSK (Lin⁻Sca1⁺c-kit⁺), SLAM LSK (Lin⁻Sca1⁺c-kit⁺CD135⁻CD48⁻CD150⁺), MPP2 (Lin⁻Sca1⁺c-kit⁺CD135⁻CD48⁺CD150⁺), MPP3 (Lin⁻Sca1⁺c-kit⁺CD135⁻CD48⁺CD150⁻), MPP4 (Lin⁻Sca1⁺c-kit⁺CD135⁺CD150⁻). Lineage (Lin) cocktail contained the following antibodies: CD19, B220, CD3, Mac1 (CD11b), Gr1 (Ly-6G/Ly-6C), CD11c, and Ter119. The following antibodies were used for flow cytometry: APC anti-mouse/human CD11b (Biolegend, 101212, M1/70, 1/250), APC anti-mouse Ly-6A/E (Sca-1) (Biolegend, 108112, D7, 1/250), APC/cyanine7 anti-mouse CD19 (Biolegend, 115529, 6D5, 1/250), brilliant violet 421 anti-mouse CD117 (c-Kit) (Biolegend, 105828, 2B8, 1/250), brilliant violet 421 anti-mouse CD3ε (Biolegend, 100341, 145-2C11, 1/250), brilliant Violet 605 anti-mouse CD150 (SLAM) (Biolegend, 115927, TC15-12F12.2, 1/1000), brilliant violet 605 anti-mouse CD19 (Biolegend, 115539, 6D5, 1/250), brilliant violet 605 anti-mouse CD45.1 (Biolegend, 110737, A20, 1/1000), brilliant violet 605 anti-mouse CD45.2 (Biolegend, 109841, 104, 1/250), eBioscience Fixable Viability Dye eFluor 780 (Thermofisher, 65-0865-14, 1/2000), FITC anti-mouse CD11c (Biolegend, 117306, N418, 1/250), FITC anti-mouse CD19 (Biolegend, 152404, 1D3/CD19, 1/250), FITC anti-mouse CD3ε (Biolegend, 100306, 145-2C11, 1/250), FITC anti-mouse CD45.2 (Biolegend, 109806, 104, 1/250), FITC anti-mouse Ly-6G/Ly-6C (Gr-1) (Biolegend, 108406, RBC6-8C5, 1/500), FITC anti-mouse TER-119 (Biolegend, 116206, TER-119, 1/250), FITC anti-mouse/human CD11b (Biolegend, 101206, M1/70, 1/250), FITC anti-mouse/human CD45R/B220 (Biolegend, 103206, RA3-6B2, 1/250), PE anti-mouse CD135 (Biolegend, 135306, A2F10, 1/500), PE anti-mouse Ly-6G/Ly-6C (Gr-1) (Biolegend, 108408, RBC6-8C5, 1/2000), PE/cyanine7 anti-mouse CD45.2 (Biolegend, 109830, 104, 1/250), and PerCP/cyanine5.5 anti-mouse CD48 (Biolegend, 103422, HM48-1, 1/500). See also Supplementary Table 1.

## Cytokine quantification

The LEGENDplex mouse inflammation panel (740150, BioLegend) was used according to the manufacturer's instructions. Briefly, after crushing the bones, the cell suspension was centrifuged at 1500 rpm for 5 min at 4 °C. The bone marrow supernatant was then collected in a separate tube and stored at −80 °C until further use. Samples were diluted 1:1 in PBS, incubated with beads conjugated with the respective antibodies, and analyzed on a Fortessa FACS analyzer (BD Biosciences). All data were analyzed using FlowJo (BD) software.

## Misfolded (polyubiquitinated) protein quantification

$2 \times 10^6$ lineage-depleted cells were stained with the antibody mix in FACS buffer for 30 min at 4 °C to identify HSCs. Cells were washed and resuspended in 1% PFA for 10 min at room temperature. Samples were

then washed and incubated overnight with 1:500 primary pan-ubiquitin antibody (clone FK2, 04-262, Sigma) in permeabilization buffer (00-8333-56, ThermoFisher Scientific). Then, cells were washed and resuspended with 1:2500 goat anti-mouse IgG1 secondary antibody in permeabilization buffer at 4 °C for at least 2 h. After washing and resuspending in FACS buffer, samples were acquired on a Fortessa FACS analyzer (BD Biosciences). All data were analyzed using FlowJo (BD) software. For measurement of misfolded proteins in G0 HSCs cells were washed and resuspended with 1:2500 goat anti-mouse IgG1 secondary antibody in permeabilization buffer at 4 °C for 2 h, along with anti-Ki67 antibody at 4 °C for at least 2 h (H3569, Life Technologies). After washing and resuspending in FACS buffer, samples were acquired on a Fortessa FACS analyzer (BD Biosciences). All data were analyzed using FlowJo (BD) software.

## Measurement of protein concentration

For total protein concentration measurement, 1000 EPCR SLAM cells were sorted from young and aged WT and *Mda5*[-/-] mice. Cells were lysed in RIPA buffer (89900, ThermoFisher) with protease inhibitor (1x) for 10 min at 4 °C. Samples were centrifuged at 13,000 × g for 15 min to pellet the cell debris. The protein concentration in the supernatant was measured by Qubit 3.

## Unfolded protein quantification

Tetraphenylethene maleimide[39,61] (TPE-MI, HY-143218, MedChemExpress) was reconstituted according to the manufacturer's instructions, diluted in PBS (50 mM final concentration), and added to each sample. $2 \times 10^6$ lineage-depleted cells were incubated for 45 min at 37 °C. Samples were washed and stained with the antibody mix in FACS buffer for 30 min at 4 °C to identify HSCs. Cells were washed and acquired on a Fortessa FACS analyzer (BD Biosciences). All data were analyzed using FlowJo (BD) software.

## Mitochondrial mass and membrane potential quantification

$2 \times 10^6$ lineage-depleted cells were resuspended in complete StemPro-34 SFM medium with either MitoTracker Green (MTG, 50 nM final concentration, M7514, ThermoFisher Scientific) or tetramethylrhodamine methyl ester perchlorate (TMRM, 100 nM final concentration, I34361, ThermoFisher Scientific), supplemented with verapamil (V4629, Sigma Aldrich). After 30 min incubation at 37 °C, samples were washed and stained with the antibody mix in FACS buffer with verapamil for 30 min at 4 °C to identify HSCs. Cells were washed and acquired on a Fortessa FACS analyzer (BD Biosciences). All data were analyzed using FlowJo (BD) software.

## Measurement of protein synthesis

$2 \times 10^6$ lineage-depleted cells were stained with the antibody mix in FACS buffer for 30 min at 4 °C to identify HSCs. Then, cells were resuspended in complete StemPro-34 SFM medium (10639011ThermoFisher) with OP-Puro at 50 μM and incubated at 37 °C for 1 h. Samples were then washed, fixed with 1% PFA in PBS for 10 min at 4 °C, and permeabilized with permeabilization buffer (00-8333-56, ThermoFisher Scientific) for 20 min at 4 °C. The azide-alkyne cycloaddition was performed using the Click-iT Plus OPP Alexa Fluor 488 Kit (C10456, ThermoFisher Scientific), according to the manufacturer's instructions. After the 30 min reaction, the cells were washed with PBS and acquired on a Fortessa FACS analyzer (BD Biosciences). All data were analyzed using FlowJo (BD) software.

## Autophagic flux detection

This assay was performed using the CYTO-ID autophagy detection kit (ENZ-KIT175-0050, ENZO Life Sciences) according to manufacturer's instructions. Briefly, $2 \times 10^6$ lineage-depleted cells were resuspended with CYTO-ID green detection reagent diluted in assay buffer and incubated for 30 min at 37 °C. Samples were then stained with the antibody mix in FACS buffer for 30 min at 4 °C to identify HSCs. After

washing, cells were resuspended in FACS buffer and acquired on a Fortessa FACS analyzer (BD Biosciences). All data were analyzed using FlowJo (BD) software.

## Targeted metabolomics

For low-input targeted metabolomics (of polar metabolites), 5000 HSCs were isolated as described above. Samples were then washed with 1 mL of 9 g/L NaCl and filtered through a 30 mM cell strainer before proceeding immediately with sorting using 2 g/L NaCl in deionized water as sheath liquid. Cells were sorted directly into a 96-well plate containing 25 μL of extraction solution, which consisted of 0.1% $^{13}C$ internal standard stock (1 aliquot of U-$^{13}C$ yeast extract (Isotopic solutions, Cat# ISO-1) in 10 mL of 25% v/v methanol in water) in acetonitrile (LC-MS grade). To assess background levels, 5000 cell debris (low light-scattering signals) were collected. Targeted metabolite quantification by LC-MS was carried out as previously described[90], and data processing was performed using the R package automRm[91]. Metabolites were only included if they were detected above the cell debris detection levels, and the retention time was identical to the $^{13}C$ yeast standard qualifier peak. The metabolic differences were calculated by determining the area under the curve (AUC). Data were normalized to the mean of every detected metabolite in the aged WT population per experiment.

## HSF1, phospho-HSF1, phospho-eIF2α, and dsRNA staining

HSCs were isolated and sorted as previous described. Cells were loaded into cytospin (700 rpm, 5 min) to allow them to adhere to glass slides. Fixation was performed with 4% PFA in PBS for 10 min at room temperature, slides were air-dried and kept at +4 °C until the next step. Permeabilization and blocking steps were done with 0.1% Triton X-100 and 10% goat serum (Sigma G9023), respectively. Indirect immunofluorescent analysis was conducted using primary anti-HSF1 antibodies (Enzo Life Sciences ADI-SPA-901-D, 1:500 or StressMarq SMC-118D for Rat Ab 1:500), anti-dsRNA (Sigma Aldrich MABE1134, 1:500), phosphorylated HSF1 (Enzo Life Sciences [pSer326]HSF1 ADI-SPA-902-D 1:500) and phosphorylated EIF2a (cell signaling phospho-EIF2a (Ser51) 9721S 1:500) and secondary anti-rabbit or anti-mouse antibodies conjugated with AlexaFluor-488 dye (ThermoFisher Scientific,1:1000). Samples were mounted with Fluroshield mounting solution containing DAPI (Sigma F6057). Images were acquired using an LSM880 confocal microscope (Carl Zeiss, Germany).

## HSF1 staining in HEK293T cells

HEK293T cells are cultured in IMDM (10% FBS, 2% penicillin, streptomycin, and glutamine). 20.000 cells were plated in each well of 12-well plates, on coverslips. pFLAG_WT MDA5 (pFLAG_WT MDA5 was a gift from Sun Hur (Addgene plasmid # 216798; http://n2t.net/addgene:216798; RRID:Addgene_216798) and pFLAG_MDA5Δ2CARD (pFLAG_MDA5Δ2CARD was a gift from Sun Hur (Addgene plasmid # 216800; http://n2t.net/addgene:216800; RRID:Addgene_216800) were transfected using Lipofectamine overnight according to the manufacturer's instructions. Fixation was performed with 4% PFA in PBS for 20 min at room temperature. Permeabilization and blocking steps were done with 0.1% Triton X-100 and 1% BSA, respectively. Indirect immunofluorescent analysis was conducted using primary anti-HSF1 antibodies (Enzo Life Sciences ADI-SPA-901-D,1:1000) and AlexaFluor-488 dye (ThermoFisher Scientific,1:1000). Samples were mounted with Fluroshield mounting solution with DAPI (Sigma F6057). Images were acquired using an LSM880 confocal microscope (Carl Zeiss, Germany).

## Proximity ligation assay

Human BJ fibroblasts, untreated or treated with 24 h at 10 μg/mL PolyI:C, were fixed with 4% PFA for 20 min at 37 °C and permeabilized with 2% Triton X-100 for 10 min at room temperature. PLA was performed according to the manufacturer's protocol (Sigma Aldrich

DUO94104). Primary antibodies were used at the following dilutions: anti-HSF1 (Santa Cruz sc-17757, 1:100), anti-MDA5 (Abcam ab79055, 1:200). Cells were mounted with DAPI-containing mounting medium (Sigma Aldrich DUO82040). Interaction between HSF1 and MDA5 was visualized in situ using a Carl Zeiss LSM880 confocal microscope (Carl Zeiss). Presented images represent the maximum intensity projection of the z-stack.

## RNA-seq library preparation

Two thousand five hundred EPCR SLAM cells were sorted from young, middle-aged, and aged WT and *Mda5*[-/-] mice. RNA was then isolated using the Arcturus PicoPure RNA isolation Kit (KIT0204, Applied Biosystems). Low-input RNA-seq libraries were prepared using SMART-Seq Stranded Kit (634442, Takara Bio). First, random priming allowed the generation of cDNA from all RNA fragments in the sample. Next, a first round of PCR amplification (5 cycles) added full-length Illumina adapters (Indexing Primer Set HT for Illumina v2), including barcodes. Then, the ribosomal cDNA (originating from rRNA) was cleaved. The cDNA fragments were enriched via a second round of PCR amplification (14 cycles) using primers universal to all libraries. Libraries were cleaned up using AMPure XP beads (A63881) following the manufacturer's protocol. Libraries were sequenced at Vanderbilt.

## RNA-seq analysis of genes

Paired-end 300 bp reads were generated using the Illumina NovaSeq 6000 system and sequenced at a depth of 48–70 M. Raw fastq read files were assessed for quality with FastQC v0.11.9 (http://www.bioinformatics.babraham.ac.uk/projects/fastqc/) and trimmed of Illumina sequencing adapters with Trim Galore v0.3.7 (https://www.bioinformatics.babraham.ac.uk/projects/trim_galore/). Alignment to the reference mouse genome (GENCODE GRCm38.p6 release M23) was done using STAR v2.7.10a[92] with default parameters. Aligned bam files were indexed with samtools v1.12[93], and gene count matrices for each sample were generated from the alignment bam files using HTSeq-count v0.12.4[94] with the parameter –intersection-non-empty, and –stranded = "reverse" was specified for RNA libraries prepared with the Takara directional kit. Samples were hierarchically and PCA clustered to determine sample congruence. Normalization, scaling, and differential expression analysis were performed in R v4.1.3 (R Core Team, 2022) using the edgeR v3.34.1[95] qlf test for each pair-wise comparison. Genes were considered significantly and differentially expressed between conditions if they met FDR and fold-change cutoffs of 0.05 and 1.5, respectively. Plotting of results was performed using heatmap.2 function of the gplots package v3.1.1 and ggplot2 v3.3.5[96] packages in R.

## RNA-seq analysis of transposable element families

Read generation and data pre-processing were performed as described above for RNA-seq analysis of genes. Alignment to the reference mouse genome (GENCODE GRCm38.p6 release M23) was done using STAR v2.7.10a[92] with the following parameters to retain multi-mapping reads: --outFilterMultimapNmax 100 --winAnchorMultimapNmax 100 --outMultimapperOrder Random --outSAMmultNmax 1. Alignment BAM files were indexed with samtools index[93]. Expression of transposable element families was quantified by sample using TEcounts v2.0.3[97] with parameter –mode = multi to estimate transposable element abundances from multimapped alignments, and –stranded reverse to properly process the stranded reads, using the pre-generated GENCODE GRCm38 TE GTF RepeatMasker file from the Hammel lab FTP site (https://labshare.cshl.edu/shares/mhammelllab/www-data/TEtranscripts/TE_GTF) and the GENCODE GRCm38 v23 annotation GTF. Counts were assembled into a single table in R, were normalized to counts per million (CPM), and differential expression was assessed in R v3.1.4 (R Core Team, 2022) on the combined transposable element /gene counts using the edgeR qlf test, as described

above, determining TEs to be differentially expressed if they met the cutoffs of FC 1.5 and FDR 0.05.

## Functional enrichment and upstream regulator analysis

Ingenuity pathway analysis (IPA) for upstream regulators and canonical pathways was performed on log-transformed differential expression values for genes found to be significantly differentially expressed in edgeR, using the default settings of IPA v.24.0.2[55].

## Gene set enrichment analysis

Normalized logCPM (counts per million) matrices for all genes of each comparison were extracted from the edgeR DGE object in R, generated as described above, and saved as a tab-separated text file for input into GSEA (v4.4.0 build 18)[98,99]. GSEA was run using the gene_set permutation with the max and min set sizes specified at 1000 and 1, respectively. Gene sets tested were obtained from MSigDB (https://www.gsea-msigdb.org/gsea/msigdb)[100] and[85,101,102].

## Single-cell RNA-seq

100,000 LSK cells from young and aged WT or *Mda5*[-/-] mice were isolated and sorted as previously described. Samples were then barcoded using the 10x Genomics 3' CellPlex Kit. Cell multiplexing was performed before sorting. Pooled gene expression and feature barcoding fastq files obtained via NovaSeq 6000 sequencing platform were demultiplexed and aligned to the GRCm38/mm10 reference genome using cellranger count[103]. The resulting gene expression, barcode, and hashtag matrices were imported into R v4.1.3, and analysis was performed using the Seurat package v4.1.0[104]. Cells were filtered for nFeature_RNA > 1000 and a mitochondrial gene percentage of at most 5%. Hashtag data was used to trace pooled cells of a sample back to individual mice via the HMOdemux function in Seurat. Of 38,137 WT aged cells output by cellranger count, 21,241 were singlets, and 12,616 cells were retained after filtering for further analysis. Of 34,017 *Mda5*[-/-] aged cells output by cellranger count, 18,248 were singlets, and 12,084 cells were retained after filtering for further analysis. Of 32,441 WT young cells output by cellranger, 18,788 were singlets, and 9986 cells were retained after filtering for further analysis. Of 34,636 *Mda5*[-/-] young cells output by cellranger count, 14,007 were singlets, and 8178 cells were retained after filtering for further analysis. Samples were individually regularized using SCTransform V2 before Seurat anchor-based integration using 2000 anchors. Based on individual gene expression results, aged sample replicates CMO303 (WT) and CMO306 (*Mda5*[-/-]) were found to be unfit for further analysis and were excluded. After PCA dimensionality reduction, an elbow plot was used to determine the optimal number of dimensions to proceed with (30), and SNN clustering was performed at several resolutions (0.1–1, increment of 0.1) for input into clustree[105] to determine the optimal clustering resolution. Differential expression between clusters was performed with the FindAllMarkers function, and cell populations were determined based on the expression of HSC and MPP markers[56]. To perform a comparison at the sample level, pseudobulk analysis was carried out by collapsing cell population clusters by condition and differential expression analysis in edgeR qlf test[95]. Functional enrichment, GSEA, and IPA analysis were performed on the results as described for RNA-seq analysis above.

## ATAC-seq library preparation

For ATAC-seq (3000–5000 cells per sample, 3 biological replicates per condition), HSCs were sorted in FACS buffer. HSCs were spun down at 500 g, 4 °C for 10 min, and incubated with Illumina TAGMENT DNA enzyme and buffer (20034210, Illumina) or with tagmentase (Diagenode, C01070012) and the respective buffer (Diagenode C01019043) and Digitonin (0.01%) for 30 min at 37 °C on a thermomixer shaking at 700 rpm. Transposed DNA was purified using MinElute Cleanup (28004, QIAGEN). Adapter indices were ligated using the Nextera XT

Index Kit (15055289, Illumina), and libraries were amplified using NEBNext High-Fidelity 2X PCR Master Mix (M0541S, NEB).

## ATAC-seq differential accessibility analysis

Paired-end 100 bp reads were generated using the Illumina Hiseq 3000 system and sequenced at a depth of 60–96 M. Fastq read files were trimmed of Nextera adapters with Trim Galore v0.6.7 (https://www.bioinformatics.babraham.ac.uk/projects/trim_galore/) and aligned to GRCm38/mm10 with Bowtie2 v2.4.4[106]. The resulting BAM files had PCR duplicates, mitochondrial reads, and ENCODE blacklist[107] features removed before further analysis using Picard v3.4.0 (Broad Institute, 2019), samtools v1.13[93], and bedtools v2.30.0[108], respectively. For differential accessibility analysis, BAM files were merged by sample using Samtools before calling peaks with MACS v2.2.9.1[109]. Induced/reduced peaks were determined by overlapping peaksets with bedtools intersect[108], after which gene TSSs overlapping peaks (−100 kb, +50 kb) were determined with bedtools closest[108], and motif enrichment was obtained with findMotifsGenome.pl script from the HOMER v5.1 package[110]. For visualization, filtered BAM files were converted to bigwig format via a bedgraph intermediate with bamCoverage and bedtobigwig, using scaling factors obtained from deeptools v3.5.6[111] multiBamSummary. These bigwigs, along with the previously obtained induced and reduced peaks in bed format, were input into deeptools computeMatrix, and the output was plotted using deeptools plotHeatmap reference-point.

## ATAC-seq digital genomic footprinting

Footprinting analysis to determine TF occupancy was performed on sample-merged bam files (samtools merge[93]) and sample-merged narrowpeak output from MACS2 using the Wellington algorithm from the pyDNase package v0.3.0[112]. Output bed regions were merged, then intersected among samples in order to plot profiles with dnase_average_profile.py.

## Reporting summary

Further information on research design is available in the Nature Portfolio Reporting Summary linked to this article.

## Data availability

The BioProject PRJNA1290859 and associated SRA metadata are available [https://www.ncbi.nlm.nih.gov/bioproject/?term=PRJNA1290859]. All data are included in the Supplementary Information or available from the authors, as are unique reagents used in this Article. The raw numbers for charts and graphs are available in the Source Data file whenever possible. Source data are provided with this paper.

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

## Acknowledgements

This work was supported by the French Government (National Research Agency, ANR) through the "investments for the future" programs LABEX SIGNALIFE ANR-11-LABX-0028-01 and IDEX UCAJedi ANR-15-IDEX-01 to ET, but also the Project ANR-23-CE14-0017 heTEro. The German Research Foundation, grant nos. 322977937/GRK2344-MeInBio and GZ TR 1478/2-1 (to E.T., P.B., and A.K.), the grant no. Fondation Recherche Médicale (FRM) AJE202010012488 (to E.T.) and an EHA bilateral collaborative grant 2022, and from Programmes labellisés - Fondation ARC 2025 (ARCPGA2024120009098_9649) to E.T. The work was also supported by the Leukemia Research Foundation (to M.D.), When Everyone Survives Foundation (to M.D.), P30 CA021765 (National Cancer Institute) (to M.D.). This work was supported by the Max Planck Society, the ERC-Stg-2017 (VitASTEM, 759206), the German Research Foundation (DFG) under the German Excellence Strategy (CIBSS-EXC-2189, Project #390939984), SFB1425 (Project #422681845; P09), SFB992 (Project #192904750; B07), SFB1479 (Project #441891347; P05), the European Union's Horizon 2020 Research and Innovation Programme under the Marie Skłodowska-Curie Actions Grant (ARCH; Project #813091) and the José Carreras Leukämie-Stiftung, DFG Research Training Group MeInBio (Project #322977937/GRK2344) all awarded to N.C-W. The authors acknowledge PICMI and Cytomed, the IRCAN's Imaging facility core part of the «Microscopie Imagerie Cytométrie Azur» (MICA) GIS IBiSA labeled

platform. PICMI AND/OR Cytomed was supported by the Association pour la Recherche sur le Cancer (ARC), by the national research infrastructures in biology and health (GIS IBiSA) as part of MICA and by the Conseil Départemental 06 et Région Sud». PICMI, IRCAN animal core facility and Cytomed was supported by FEDER, Contrat Plan Etat Région (CPER), Ministère de l'Enseignement Supérieur et de la Recherche, Région Sud, Conseil Départemental 06, ITMO Cancer Aviesan (Plan Cancer), Cancéropole PACA, INCa, CNRS and Inserm. This work is part of V.B. and P.B.'s PhD thesis project at Albert-Ludwigs-University Freiburg and G.T.V. thesis at the University of Côte d'Azur conducted toward the Doctor of Biology degree. The authors would like to thank all the facilities at MPI-IE and IRCAN.

## Author contributions

V.B. and G.T.V. designed and performed the majority of experiments. P.B. and A.P. developed and performed the computational analysis (bulk and single cell RNA-seq for P.B. and ATAC-seq for A.P.). M.D., L.T., and N.M. performed the HSC competitive transplants. M.-E.L. helped with the metabolomics experiments. B.G. did the PLA experiments. W.D. and T.M. helped with bioinformatic analysis. F.B. gave microscopy advice. S. performed the scRNA-seq experiments. H.M. provided cells for the study. D.V.B., N.C.W., M.D., and E.T. supervised the study and gave suggestions to the experimentalists. E.T. wrote the manuscript with input from all the authors. All the authors have read and approved this manuscript.

## Competing interests

The authors declare no competing interests.
