## [Transparent Peer Review File · Nature Communications]

Lack of MDA5 delays hematopoietic aging by modulating inflammaging and proteostasis in mice

Corresponding Author: Dr Eirini Trompouki

Version 0:

Reviewer comments:

Reviewer #2

(Remarks to the Author)

The authors present a large body of work that convincingly show that Mda5 plays a role in HSC function. The changes in inflammatory cytokine levels, improved MMP, and a role for mitigating MDA5^{-/-} to improve many aspects of HSC aging. Please find suggestions / clarifications requested to improve the overall message of the manuscript.

Suggestions / clarifications

The changes in chromatin accessibility between the young and old WT and the old MDA5^{-/-} vs old WT are particularly relevant. The overall changes (from interpretation of the figure 1) in the chromatin accessibility between the KO and WT during aging suggest that, in general, the age associated increase in accessibility is maintained in both the WT and KO HSCs, but the wording that “the young Mda5^{-/-} HSCs have more accessible chromatin, but this is reversed during aging, where 11.904 sites were induced in WT and 5.336 sites were induced in Mda5^{-/-} HSCs” is a bit confusing as the old Mda5^{-/-} HSCs have more accessible chromatin compared to the young Mda5^{-/-} HSCs. Were the same sites that were more open in young Mda5^{-/-} HSCs the ones that became closed in old Mda5^{-/-}? Could the authors please explain in Fig 1E, why the reduced peaks in aged MDA5^{-/-} (blue line plot) is higher relative to the peak in the aged WT? It is unclear if the Mda5^{-/-} aged HSCs revert chromatin accessibility to something more similar to young WT (a venn diagram or upset plot could help clarify this) as the pathway enrichments have very similar reduced and induced pathways- this may be clarified a bit with showing the overlapping loci or a more clear depiction of if the aged Mda5^{-/-} HSCs have chromatin accessibility more similar to young or just different from aged WT HSCs (which may be the interpretation from the clustering in Supp 1H)

With the significant enrichment of open chromatin in intergenic regions in the Mda5^{-/-} young HSCs- if just comparing the TE expression between young WT HSC and Mda5^{-/-} HSCs is there an increase in TE expression?

This single cell data is of course a nice additional for examining cellular heterogeneity, but the presentation of data does not follow with the manuscript- it is unclear how the Mda5^{-/-} cells differ from their WT counterparts and the pseudotime trajectories include both Mda5^{-/-} and WT cells in the young to old comparisons (and in the supplemental files show the young and old KO and young and old WT, but not the matching aged WT and KO, which appear to be the most relevant. Were there differences in the pseudotime projections of the old KO vs old WT (or perhaps unclear labels on plots)

Minor points:

1. For the RNAseq clustering for PCA, it is a bit unusual that the young HSC (both the MDA^{-/-} and WT) samples are the most variable compared to middle and old aged samples- as it is also unclear what the axis is- as the BVC: Bockfjorden volcanic complex is not a term I can find relative to principal component assays.
2. Authors nicely include middle aged HSCs in ATAC seq data, but in clustering analysis these are not included (Supp Fig 1H). please do
3. In evaluation of mitochondria- the authors measure mitochondrial mass, where they normalize to fold change relative to aged WT, yet the aged WT bar plot is not at 1. (fig 2e) this is the same for analysis of protein in figure Sup 7a/b.
4. One potential explanation for the significantly increased chimerism of the secondary transplants of the young Mda5^{-/-} that should be at least discussed is the increase in the frequency of the LSKs and HSCs in the primary transplant (not significant

due to the transplanted animals with very low chimerism).

Reviewer #3

(Remarks to the Author)

In this study, the authors demonstrate that MDA5, an RNA sensor in innate immunity, is involved in the aging of haematopoietic stem cells (HSCs). The revised manuscript incorporates updated data analyses and new experimental results, strengthening the claim that MDA5 contributes to the regulation of HSC ageing. The reviewer acknowledges that the authors have responded to the previous comments.

However, regarding comment (3), the authors provide new data using MDA5 mutants in HEK293 cells (Supplementary Figure 7G). These results indicate that the MDA5 G496R mutant, a gain-of-function variant, fails to retain HSF1 in the cytoplasm. This observation may contradict the authors' assertion that "signaling activity of MDA5 is essential for its interaction with HSF1." Furthermore, the authors include proximity ligase assay data using human BJ fibroblasts (Figure 8H); however, the nuclear detection of MDA5 remains unclear and requires further clarification. These issues should be addressed.

Minor points:

1. On page 9, line 2, the manuscript states: "Ingenuity pathway analysis in genes deregulated between aged WT and Mda5^{-/-} HSCs showed enrichment for "eukaryotic translation elongation", "EIF2 signaling", and "cellular response to heat stress" among others (Figure 4E)", and on line 12: "We then investigated the transcriptomic signature of "cellular response to heat stress" (Figure 4E)." However, the description of "cellular response to heat stress" is missing from Figure 4E.
2. The description of the mice from which the cells were harvested is absent in Supplementary Figure 6E.
3. Spelling out certain technical terms would make the manuscript more accessible to a broader readership; for example: EIF2a, MPP, cGAS.

Reviewer #4

(Remarks to the Author)

It appears that the concerns of reviewer 1 were sufficiently addressed in the revised version of the manuscript. The authors added a large amount of additional experimental data, revised the bioinformatic analysis of obtained data and edited the text to make it more accessible for readers.

Reviewer 1 was particularly concerned whether MDA5 deficient cells are inherently more fit rather than able to maintain proteostasis during aging. To address this point, the authors have added several new data sets in the main text and the supplement. Taken together, these data sets point to altered proteostasis in the absence of MDA5 beginning at middle age – in line with their main conclusion. In addition, a more extensive discussion of this aspect has been included.

In response to reviewer 1, the authors now performed ATAC-seq experiments, repeated RNA-seq experiments with freshly isolated HSCs, and sequenced all samples together to minimize batch effects. Furthermore, the bioinformatic analysis has been substantially revised, now including Ingenuity Pathway Analysis (IPA) and Gene Set Enrichment Analysis (GSEA). The analysis further supports the main conclusion of the manuscript.

Immunofluorescence microscopy was substantially extended and now includes the use of new antibodies and experimental approaches (proximity ligation) to verify a role of MDA5 in controlling the location and activation of HSF-1.

Finally, additional experiments were performed to analyze the impact of MDA5 absence on protein synthesis and the ubiquitylation and aggregation of proteins at different age levels of HSCs and MPPs. This provides further evidence for a participation of the innate immune RNA sensor MDA5 in the regulation of proteostasis.

Version 1:

Reviewer comments:

Reviewer #2

(Remarks to the Author)

Thank you for the clarifications and additional information. I agree the venn diagram overlap is not needed for the manuscript, and I may have been unclear about the comparisons I was interested in: I was curious about the overlap of aged MDA^{-/-} more closed compared to aged WT with the chromatin changes in young vs old WT: However, the authors do state now that the aged MDA5^{-/-} HSCs do not resemble more young HSCs, answering the question.

All other comments / suggestions were appropriately addressed- Please do double check through the newly added text as there were some minor typos- but the science looks interesting and well presented

Reviewer #3

(Remarks to the Author)

The reviewers acknowledge that the authors have addressed the comments and strongly commend this research for its contribution to the advancement of the field.

>We kindly disagree with the reviewer since the WT and Mda5^{-/-} young and aged HSCs are clearly depicted in the Supplementary Figure 6E.

My apologies. My previous comment was incorrect. The reviewer intended to refer to “Supplementary Figure 7E”, not “6E”. In Supplementary Figure 7E, it would be helpful for readers if the three panels were labelled to indicate which cells originate from the ‘young’, ‘middle’, and ‘aged’ groups.

Reviewer #2 (Remarks to the Author):

The authors present a large body of work that convincingly show that Mda5 plays a role in HSC function. The changes in inflammatory cytokine levels, improved MMP, and a role for mitigating MDA5^{-/-} to improve many aspects of HSC aging. Please find suggestions / clarifications requested to improve the overall message of the manuscript.

We thank the reviewer for the kind words.

Suggestions / clarifications

The changes in chromatin accessibility between the young and old WT and the old MDA5^{-/-} vs old WT are particularly relevant. The overall changes (from interpretation of the figure 1) in the chromatin accessibility between the KO and WT during aging suggest that, in general, the age associated increase in accessibility is maintained in both the WT and KO HSCs, but the wording that “the young Mda5^{-/-} HSCs have more accessible chromatin, but this is reversed during aging, where 11.904 sites were induced in WT and 5.336 sites were induced in Mda5^{-/-} HSCs” is a bit confusing as the old Mda5^{-/-} HSCs have more accessible chromatin compared to the young Mda5^{-/-}-HSCs.

We agree with the reviewer and we have reworded this phrase to “In the comparison of the young WT and *Mda5*^{-/-} HSCs, we found 1.723 regions that were mostly accessible in WT (reduced peaks), and 10.694 mostly accessible in *Mda5*^{-/-} HSCs (induced peaks) (**Figure 1D and Supplementary Table 1**). Aged *Mda5*^{-/-} HSCs on the other hand have less accessible chromatin than the WT, with 11.904 sites induced in WT and 5.336 sites induced in *Mda5*^{-/-} HSCs (**Figure 1E and Supplementary Table 1**).”

Were the same sites that were more open in young Mda5^{-/-} HSCs the ones that became closed in old Mda5^{-/-}?

Only 491 common regions are open in Young *Mda5*^{-/-} and closed in Aged *Mda5*^{-/-} (for this reason we only have the venn as an answer to the reviewer but we can add it to the text if you think it is helpful) .

Could the authors please explain in Fig 1E, why the reduced peaks in aged MDA5-/- (blue line plot) is higher relative to the peak in the aged WT?

The reviewer is correct and there was a problem with the normalization of this sample. Now this problem is addressed.

It is unclear if the *Mda5*^{-/-} aged HSCs revert chromatin accessibility to something more similar to young WT (a venn diagram or upset plot could help clarify this) as the pathway enrichments have very similar reduced and induced pathways- this may be clarified a bit with showing the overlapping loci or a more clear depiction of if the aged *Mda5*^{-/-} HSCs have chromatin accessibility more similar to young or just different from aged WT HSCs (which may be the interpretation from the clustering in Supp 1H)

As you can see from the new plot on Figure 1H the majority of the open regions in aged HSCs are common between the aged WT and the aged *Mda5*^{-/-}. However 11779 are only open in aged WT HSCs and 4460 regions are open only in aged *Mda5*^{-/-} HSCs, in comparison to all narrow peaks from all datasets. Very few genomic regions are common with the young HSCs.

With the significant enrichment of open chromatin in intergenic regions in the *Mda5*^{-/-} young HSCs- if just comparing the TE expression between young WT HSC and *Mda5*^{-/-} HSCs is there an increase in TE expression?

Even if some genomic regions are more open in the *Mda5*^{-/-} the TE expression does not reflect this. There is no significantly deregulated TE family between young or middle aged *Mda5*^{-/-} versus young or middle aged WT HSCs and only one family is upregulated in aged *Mda5*^{-/-} versus aged WT HSCs (See also Supplementary Table 3).

This single cell data is of course a nice additional for examining cellular heterogeneity, but the presentation of data does not follow with the manuscript- it is unclear how the *Mda5*^{-/-} cells differ from their WT counterparts and the pseudotime trajectories include both *Mda5*^{-/-} and WT cells in the young to old comparisons (and in the supplemental files show the young and old KO and young and old WT, but not the matching aged WT and KO, which appear to be the most relevant. Were there differences in the pseudotime projections of the old KO vs old WT (or perhaps unclear labels on plots)

The reviewer is right and we did the pseudotime trajectories separately for each population without observing any significant difference between WT and MDA5 KO especially when aged. We didn't include this in the manuscript because we think it doesn't add much to our conclusions but we are happy to do that if the reviewer feels it is necessary.

Minor

points:

1. For the RNAseq clustering for PCA, it is a bit unusual that the young HSC (both the MDA^{-/-} and WT) samples are the most variable compared to middle and old aged samples- as it is also unclear what the axis is- as the BVC: Bockfjorden volcanic complex is not a term I can find relative to principal component assays.

We are sorry for the confusion. Indeed the reviewer is correct and we should have less variation in the young HSCs but this is not what we are observing. Regarding the BCV is biological coefficient of variation (BCV) which is a term associated with PCA. We have now explained the term in the figure legend.

2. Authors nicely include middle aged HSCs in ATAC seq data, but in clustering analysis these are not included (Supp Fig 1H). please do

We are sorry if we confused the reviewer but we have not performed the ATAC-seq on middle aged animals and that is why we have not included it in the clustering analysis.

3. In evaluation of mitochondria- the authors measure mitochondrial mass, where they normalize to fold change relative to aged WT, yet the aged WT bar plot is not at 1. (fig 2e) this is the same for analysis of protein in figure Sup 7a/b.

We have corrected the graphs and thank you very much for the thorough comments.

4. One potential explanation for the significantly increased chimerism of the secondary transplants of the young Mda5^{-/-} that should be at least discussed is the increase in the

frequency of the LSKs and HSCs in the primary transplant (not significant due to the transplanted animals with very low chimerism).

We thank the reviewer for the comment and we have added this explanation to the manuscript.

Reviewer #3 (Remarks to the Author):

In this study, the authors demonstrate that MDA5, an RNA sensor in innate immunity, is involved in the aging of haematopoietic stem cells (HSCs). The revised manuscript incorporates updated data analyses and new experimental results, strengthening the claim that MDA5 contributes to the regulation of HSC ageing. The reviewer acknowledges that the authors have responded to the previous comments.

We thank the reviewer for the insightful comments.

However, regarding comment (3), the authors provide new data using MDA5 mutants in HEK293 cells (Supplementary Figure 7G). These results indicate that the MDA5 G496R mutant, a gain-of-function variant, fails to retain HSF1 in the cytoplasm. This observation may contradict the authors' assertion that "signaling activity of MDA5 is essential for its interaction with HSF1."

We repeated this experiment multiple times and we lowered the dose of the constitutive active mutant. However all the times we got similar results and we think that the constitutively active mutant is causing too much inflammation and thus devastating to the cells. Since this is an ambiguous results we decided to remove the constitutively active mutant from our results and if the reviewer agrees.

Furthermore, the authors include proximity ligase assay data using human BJ fibroblasts (Figure 8H); however, the nuclear detection of MDA5 remains unclear and requires further clarification. These issues should be addressed.

We agree with the reviewer and we were equally surprised about the localization of MDA5 that is also nuclear and not only cytoplasmic. For this reason we isolated mouse dermal fibroblasts for WT and *Mda5*^{-/-} embryos and stained them with the MDA5 antibody under homeostatic conditions and after PolyIC stimulation that increases the levels of MDA5. We indeed show as you can see in the images below that our antibody is specific and that MDA5 is localized both in the cytoplasm and the nucleus. This is a novel observation that favors better investigation in the future but we added a comment in our manuscript. We don't think that we should include these controls in the manuscript but we are happy to do so if necessary. As a note, we would like to mention that also RIG-I has been found in the nucleus (PMID: [30097581](https://pubmed.ncbi.nlm.nih.gov/30097581/)). In the figure MDA5 is stained in red and in blue we have DAPI staining.

Minor

points:

1. On page 9, line 2, the manuscript states: ‘Ingenuity pathway analysis in genes deregulated between aged WT and *Mda5*^{-/-} HSCs showed enrichment for “eukaryotic translation elongation”, “EIF2 signaling”, and “cellular response to heat stress” among others (Figure 4E)’, and on line 12: “We then investigated the transcriptomic signature of “cellular response to heat stress” (Figure 4E).” However, the description of “cellular response to heat stress” is missing from Figure 4E.

The reviewer is right and the cellular response to heat stress was enriched but not significantly and thus we removed it from the figure and now from the text.

2. The description of the mice from which the cells were harvested is absent in Supplementary Figure 6E.

We kindly disagree with the reviewer since the WT and *Mda5*^{-/-} young and aged HSCs are clearly depicted in the Supplementary Figure 6E.

3. Spelling out certain technical terms would make the manuscript more accessible to a broader readership; for example: EIF2a, MPP, cGAS.

The reviewer is right and we have explained these abbreviations.

Reviewer #4 (Remarks to the Author):

It appears that the concerns of reviewer 1 were sufficiently addressed in the revised version of the manuscript. The authors added a large amount of additional experimental data, revised the bioinformatic analysis of obtained data and edited the text to make it more accessible for readers. Reviewer 1 was particularly concerned whether MDA5 deficient cells are inherently more fit rather than able to maintain proteostasis during aging. To address this point, the authors have added several new data sets in the main text and the supplement. Taken together, these data sets point to altered proteostasis in the absence of MDA5 beginning at middle age – in line with their main conclusion. In addition, a more extensive discussion of this aspect has been included. In response to reviewer 1, the authors now performed ATAC-seq experiments, repeated RNA-seq experiments with freshly isolated HSCs, and sequenced all samples together to minimize batch effects. Furthermore, the bioinformatic analysis has been substantially revised, now including Ingenuity Pathway Analysis (IPA) and Gene Set Enrichment Analysis (GSEA). The analysis further supports the main conclusion of the manuscript. Immunofluorescence microscopy was substantially extended and now includes the use of new antibodies and experimental approaches (proximity ligation) to verify a role of MDA5 in controlling the location and activation of HSF-1. Finally, additional experiments were performed to analyze the impact of MDA5 absence on protein synthesis and the ubiquitylation and aggregation of proteins at different age levels of HSCs and MPPs. This provides further evidence for a participation of the innate immune RNA sensor MDA5 in the regulation of proteostasis.

We thank the reviewer for thoroughly reading our manuscript.

Reviewer #2 (Remarks to the Author):

Thank you for the clarifications and additional information. I agree the venn diagram overlap is not needed for the manuscript, and I may have been unclear about the comparisons I was interested in: I was curious about the overlap of aged MDA^{-/-} more closed compared to aged WT with the chromatin changes in young vs old WT: However, the authors do state now that the aged MDA5^{-/-} HSCs do not resemble more young HSCs, answering the question.

All other comments / suggestions were appropriately addressed- Please do double check through the newly added text as there were some minor typos- but the science looks interesting and well presented

We thank the reviewer for all the useful comments.

Reviewer #3 (Remarks to the Author):

The reviewers acknowledge that the authors have addressed the comments and strongly commend this research for its contribution to the advancement of the field.

>We kindly disagree with the reviewer since the WT and Mda5^{-/-} young and aged HSCs are clearly depicted in the Supplementary Figure 6E.

My apologies. My previous comment was incorrect. The reviewer intended to refer to “Supplementary Figure 7E”, not “6E”. In Supplementary Figure 7E, it would be helpful for readers if the three panels were labelled to indicate which cells originate from the ‘young’, “middle”, and ‘aged’ groups.

We thank the reviewer for the useful comments and we have corrected the figure according to the reviewer’s instructions.